# Rationale, conceptual issues, and resultant protocol for a mixed methods Person Trade Off (PTO) and qualitative study to estimate and understand the relative value of gains in health for children and young people compared to adults

Tessa Peasgood[1,2]*, Cate Bailey[1], Gang Chen[3], Ashwini De Silva[1], Udeni De Silva Perera[3], Richard Norman[4], Koonal Shah[5], Rosalie Viney[6], Nancy Devlin[1]

1 Health Economics Unit, Centre for Health Policy, Melbourne School of Population and Global Health, University of Melbourne, Melbourne, Australia, 2 Division of Population Health, School of Medicine and Population Health, University of Sheffield, Sheffield, United Kingdom, 3 Centre for Health Economics, Monash University, Melbourne, Australia, 4 School of Population Health, Curtin University, Perth, Australia, 5 National Institute for Health and Care Excellence, London, United Kingdom, 6 Faculty of Health, Centre for Health Economics, Research and Evaluation (CHERE), University of Technology Sydney, Ultimo, Australia

* T.Peasgood@sheffield.ac.uk

## Abstract

### Background

Economic evaluation of healthcare typically assumes that an identical health gain to different patients has the same social value. There is some evidence that the public may give greater value to gains for children and young people, although this evidence is not always consistent. We present a mixed methods study protocol where we aim to explore public preferences regarding health gains to children and young people relative to adults, in an Australian setting.

### Methods

This study is a Person Trade Off (PTO) choice experiment that incorporates qualitative components. Within the PTO questions, respondents will be asked to choose between treating different groups of patients that may differ in terms of patient characteristics and group size. PTO questions will be included in an online survey to explore respondent views on the relative value of health gains to different age groups in terms of extending life and improving different aspects of quality of life. The survey will also contain attitudinal questions to help understand the impact of question style upon reported preferences. Additionally, the study will test the impact of forcing respondents to express a preference between two groups compared with allowing them to report that the two groups are equivalent. One-to-one 'think aloud', semi-structured interviews will be conducted to explore a sub-sample of respondents' motivations and views in more detail. Focus groups will be conducted with members

**Data Availability Statement:** No datasets were generated or analysed during the current study which is just the protocol. All relevant data from this study will be made available upon study completion.

**Funding:** ND, TP, CB, RN, RV, GC, UDS - Medical Research Future Fund (MRFF) Grant number APP1200816 https://www.health.gov.au/our-work/medical-research-future-fund ADS EuroQol Research Foundation grant number 348-PHD https://euroqol.org/ The funders did not and will not have a role in study design, data collection and analysis, decision to publish, or preparation of the manuscript.

**Competing interests:** I have read the journal's policy and the authors of this manuscript have the following competing interests: [ND, RV, RN, KS and TP are all members of the EuroQol group. This does not alter our adherence to PLOS ONE policies on sharing data and materials]

of the public to discuss the study findings and explore their views on the role of public preferences in health care prioritisation based on patient age.

## Discussion

Our planned study will provide valuable information to healthcare decision makers in Australia who may need to decide whether to pay more for health gains for children and young people compared with adults. Additionally, the methodological test of forcing respondent choice or allowing them to express equivalence will contribute towards developing best practice methods in PTO studies. The rationale for and advantages of the study approach and potential limitations are discussed in the protocol.

## 1. Introduction

Health Technology Appraisal (HTA) is a systematic process by which new technologies or health care interventions are assessed and prioritised against existing interventions [1]. HTA aims to provide information to policy makers on the medical, social, economic and ethical issues relating to the future use of a health technology. Agencies that conduct HTA (such as Australia's Pharmaceutical Benefits Advisory Committee (PBAC), England's National Institute for Health and Care Excellence (NICE), and the Canadian Agency for Drugs and Technologies in Health (CADTH)) have an interest in evidence, not just on the expected health outcomes of technologies, but also on the relative importance to society of those health outcomes. Whilst the default position in economic evaluation of health is often that a Quality Adjusted Life Year (QALY) has the same value regardless of who receives it [2], this assumption is not necessary and the social value of a QALY may vary with patient characteristics such as age, condition severity, health-related lifestyle and social role [2]. Funding decisions may consider the characteristics of patients receiving health gains, either explicitly through weighting gains to some patients more heavily than others [3,4] or implicitly through taking patient characteristics into consideration during deliberation [5–8].

Age has been proposed as a potential patient characteristic that should be taken into consideration [9,10]. A recent systematic review synthesized the international evidence on societal views on the relative social value of child versus adult health gains [11]. This review identified outstanding uncertainties in the relative social value of health gains across the full infant to older adolescent age range, inconsistencies in the findings based on research method, and a lack of understanding of the drivers behind these inconsistencies and of individuals' motivations when responding to stated preference questions relating to relative social value. A better understanding of the public's views relating to their willingness (or not) to prioritise child and adolescent health gains would provide timely and important information to support HTA deliberations.

This study seeks to provide evidence to decision makers in Australia on public opinion regarding the social value of child health gains relative to adult health gains. Specifically, we aim to provide (a) an estimate of the average relative weight for child health gains relative to adult health gains as judged by the Australian general public, (b) an understanding of the variation in preferences underpinning that average, and the reasons for any differences in relative value and, (c) an understanding of whether priority for child health depends on whether gains come from extensions in life years or improvements in quality of life. This is a mixed methods

study with study aims to be met by both quantitative and qualitative methods as shown in Box 1.

## Box 1. Study aims.

### Quantitative PTO survey aims

1. **Provide weights for the social value** of improvements in both length of life and health-related quality of life for each age from birth to young adulthood (0-24) versus gains to an adult (aged 40 or 55), based on the stated preferences of the Australian public.

2. **Test for difference** in child vs adult weights between:

   a. extending length of life and improving quality of life

   b. improving different domains of quality of life (physical health (pain and mobility) and mental health (distress/low mood and anxiety)

   c. extending length of life for 2 years versus 5 years

   d. including an 'opt out' equivalence or no preference option compared to forcing respondents to select a preferred age group in each question

3. Compare preferences derived from trade-off questions against attitudinal questions and show the percentage of respondents classified as potentially inconsistent.

4. Explore the internal consistency of individual responses through a 'chaining' test of preferences.

5. Explore the robustness of average responses through using bootstrapping to estimate confidence intervals.

### Qualitative aims

1. Provide a summary of the views of a sample of the Australian public and of parents with children with a health condition to understand how they feel about valuing health gains differently based on youth and for different types of health gain.

2. Provide a summary of explanations which may help shed light on any inconsistencies between attitudinal questions and PTO responses.

3. Provide a summary of views of a sample of the Australian public, including young people, parents with young children and adults without young children on how they feel about this research being used to inform decision makers.

The purpose of this protocol paper is to set out the rationale for undertaking this work; and to provide a detailed account of the options and issues which were considered in making choices about specific aspects of the study design, and to provide a full description of the study protocol.

## 2. Methods

### 2.1 Choice of study methodology

Many different approaches have been used to elicit public preferences for health care [12]. The three most common approaches used for estimating the relative social value of health gains across different groups of patients are i) Willingness to pay using contingent valuation (WTP) ii) Discrete Choice Experiments (DCE) and iii) Matching or equivalence studies. These approaches are discussed below, with specific consideration of the usability of their findings from an Australian decision maker perspective.

**2.1.1 Willingness to pay.**   Empirical work using willingness to pay (WTP) has considered the relationship between age and the social value of life. This body of work has included both revealed preference studies based on purchases of products with safety features such as vehicles [13] and stated preference (or contingent valuation) studies asking respondents' willingness to pay for mortality risk reductions [14] or morbidity reductions [15].

Willingness to pay is grounded in welfare economics theory and has been argued to be the appropriate method of assessing individual benefit from health interventions [16]. However, contingent valuation is less suited to exploring societal preferences required to support decision making within a publicly funded health care system [17]. Some researchers have aimed to incorporate a society perspective within WTP, combing self-interest alongside what is judged to be best for society [18]. For example, Reckers-Droog et al [19] use a WTP approach based on a hypothetical increase to monthly basic health-insurance premium which could be used to treat patients of different ages, for whom friends, family and the respondent themselves may belong [19]. Richardson et al (2014) developed a Relative Social Willingness to Pay approach in which respondents divide a health care budget in accordance with the perception of the relative social value of the options presented. The authors note that this approach has the advantage of using a metric that responders are familiar with (i.e. money), hence reducing cognitive complexity, whilst also making the opportunity cost within decisions apparent [20].

**2.1.2 Discrete choice experiments.**   Discrete Choice Experiments are a standard method for eliciting preferences over hypothetical scenarios containing multiple attributes and provide an estimate of the relative importance of each attribute [21]. The method has been used successfully to derive weights in this context [22].

There are many aspects of a health care decision that the public may care about and consider relevant to prioritisation of health care, including the characteristics of the treatment profile and characteristics of the individual treatment recipient. Relevant characteristics of the treatment profile include: Quality of life and length of life profile with and without treatment; Expected age of death with and without treatment; With/without treatment profiles combined to give QALY gain from treatment (disaggregated to years of life gained and quality of life gained), Burden of the health condition–i.e. expected quality of life and life expectancy given their condition and how this differs to population norms; Closeness to death; Availability of other treatment options. At the individual level, the recipient may be a patient or carer, they may currently or in the past have engaged in health harming behaviours, they may be (or have been, or will be) productively employed, they may be disadvantaged in non-health aspects (such as income). Individuals will fall into more than one group, and it is unclear how any weights could be combined where other considerations are not part of the same study. Clearly,

it is challenging to consider everything. However, it may be particularly useful for decision makers to understand the strength of any preference to weight health gains based on patient age compared with the strength of any preference for weighting health gains based on other important (non-age based) individual or treatment characteristics. A DCE approach has the potential to investigate the relative strength of these different potential modifiers, however, there are specific limitations associated with use of DCE in this context:

i. It is difficult to present decisions with clarity and to know how respondents perceive them, given the interactions with age and other potential attributes such as life expectancy or health/life experienced to date.

ii. It is challenging to probe the reasons behind reported preferences towards age weighting in a qualitative interview when many factors vary within the choice set and the focus of the discussion may move away from age.

iii. It is difficult to assess individual-level consistency between DCE choice-based questions and other types of questions, such as attitudinal questions, since within a DCE individual age-weights are not able to be estimated.

**2.1.3 Matching or equivalence studies.** A matching or equivalizing groups approach, originally called the Equivalence of Numbers [23], involves asking individuals how many outcomes of one kind are equivalent in terms of social value to a given number of outcomes of another kind [12]. This category of approaches includes the Person Trade Off (PTO) when equivalizing the numbers of patients benefiting [24] and Gain Trade Off (GTO) when equivalizing the amount of health gain (e.g., Busschbach et al, 1993 [25]; Rodriguez & Pinto, 2000 [26]).

PTO has been widely used to estimate social value weights for health gains both in the context of health state valuation [27] and to estimate social value across different groups and treatment characteristics (e.g. Baker et al., 2010 [28], Nord et al, 1996 [29] and Petrou et al 2013 [9]). In PTO questions designed to elicit preferences towards treating patients of different ages, participants make choices between pairs of (hypothetical) interventions that benefit patients from different age categories.

In GTO studies, rather than the patient group size changing, the size of the health benefit given to patients with different characteristics varies until equivalence. For example, one year of additional life granted to a 10-year-old might be judged equivalent to 2 years of life granted to a 50-year-old.

Studies may include variation in other attributes within the equivalence choice set such as cost of treatment or treatment benefit e.g. Bourke et al., 2018 [30]. Four potential concerns have been raised relating to matching approaches:

i. **Many studies have not framed the questions to focus on a marginal health gain.** Mæstada and Norheim [31] note that although valuation of marginal health gains is preferred for distribution weights, eliciting preferences based on marginal health gain is cognitively difficult, and particularly problematic with GTO as the size of the gain differs between the two groups making one group closer to a marginal gain. This suggests the PTO questions, with a close to marginal health gain, would be preferable. There may be benefit in including questions using both GTO and PTO approaches to enable a comparison; however, switching between patient group size differing and the size of the health gain differing between programs would generate significant additional complexity.

ii. **Combining gains to different groups assumes gains are additively separable.** Mæstada and Norheim [31] also raise a concern with the PTO approach in that aggregating gains to separate individuals and comparing this to gains for fewer individuals is problematic if,

morally, aggregation of gain to many individuals cannot compensate for the loss of a gain to fewer individuals. How respondents make decisions, and how they compare gains within PTO choices is important to understand. Some studies have found conflicting evidence when asking general questions about priority setting compared with PTO questions (e.g. Nord et al, 1996) [29], and deviations between ranking and PTO responses [9]. For example, Rowen et al (2016) [32] found greater support for End of Life premium with DCE results than reflected in attitudinal questions. Shah et al., (2015) [33] found some disagreement between stated preference choices and respondent agreement with researcher policy interpretation of those choices. McHugh et al. [34] raise the potential for separating preferences to different levels "principles, policies and choices" with differing levels of abstraction and specificity which might lead to different implications.

iii. **Matching questions with only one varying patient characteristic creates a focusing of attention on that characteristic**. Some concern has also been expressed that focusing on age as the only attribute which varies between choice sets may make age seem more salient than it would otherwise be–hence greater weight may be given to this attribute than would be the case were it to be placed alongside the many other potentially relevant characteristics of the recipient or the nature of health gain [35].

iv. **Extreme values.** Some PTO studies report very skewed data (e.g. Jelmsa et al, 2002 [36]; Reckers-Droog et al, 2019 [37]). A PTO design in which the most preferred group is reduced in size in order to move towards the point of equivalence creates a lower bound for the group size (i.e. 1 patient) can counter this to some extent. Respondents who always prefer one group regardless of group size do not have a trade-off ratio–they will always prefer to treat the same group no matter how small the relative gain. There is some uncertainty on how these lexicographic preferences should be treated. One option may be to report them separately and not attempt to combine them with preferences displaying trade-offs. If a limited number of iterative questions are asked to identify equivalence it is not possible to distinguish between lack of willingness to trade versus a very strong preference for one group. One approach adopted in such studies is to assume a small positive willingness to trade. For example, if respondents express a preference for the same group through to the last iterative question, they are assigned an equivalence value between zero and the previous question's preferred group size. This has the advantage that the ratio of means can be calculated including all respondents. In some cases these absolute preferences have been perceived as evidence of failure to comprehend the task or misalignment to the QALY model and dropped from the analysis (e.g. Busschbach et al, 1993 [25]).

## 2.2 Adopted study approach and justification

Notwithstanding the limitations of matching studies noted in the previous section, after careful consideration of the alternatives, a standard PTO choice-experiment incorporating deliberation and discussion was elected as our principal method. The reasons for its advantage in this context are detailed below.

The approach adopted aims to isolate a 'pure' youth effect from other aspects of value, as other attributes are held constant between the choices (e.g. the health gain is always identical between choices). This approach places the primary trade off we are interested in as the focus of the respondents' attention. Whilst we acknowledge other treatment and patient characteristics may be relevant to the social value attributed to the health gain incorporating all these would remove the focus from our question of interest.

The questions present a simpler choice for participants than a DCE in which multiple aspects of the program, treatment or patient could be used as attributes. The simpler PTO choice will help the qualitative components of the study to remain focused on the issue of patient age.

In this study, we wish to capture the views of members of the public of all age ranges. This is important because each age group may feel differently about prioritising patients of their own age. We considered participants' age of 16 years to be an appropriate benchmark in terms of maturity, literacy and comprehension. At 16 years, participants should have the ability to both consent to the research and have the capacity to understand the questions relating to healthcare prioritising. The desire to include adolescents (16–18 years) in the study further influenced our preference for the simplest style of question used in PTO.

A further advantage of PTO questions is the ability to include individual level comparisons between different types of questions into a survey. Respondents' inconsistencies can also be discussed with them in the qualitative component.

Two qualitative components will complement the quantitative data collection, one based on an extended cognitive debrief/'think aloud' approach in one-to-one interviews (about 40 interviews) and one based on focus groups with deliberation (about 4 focus groups). In both cases these will be with members of the public (the details of these samples are discussed in 2.6 below).

In the interviews, reasons behind respondents' choices will be discussed, and the PTO style question will enable the interviewer to feed back the respondent's relative weights for improving children's health relative to adults between different questions, and highlight any differences based on context (e.g. different aspects of quality of life). The interviewer will also introduce other opinions expressed by participants in the study and related research to explore the respondent's reaction to alternative viewpoints. Participants will be encouraged to discuss whether they feel differently about their answers given in the survey after the discussion.

The reasons why people might give preference to treating patients of different ages are multiple and complex [11]. Incorporating a qualitative and deliberative component is important to help understand these reasons, and to delve behind respondents' initial answers to understand the robustness of their preferences. A preference for one age group may arise from a pure age effect (such as a 'fair innings' judgement [10] or willingness to prioritise due to childhood vulnerability) but they may also arise because of judgements about different capacity to benefit from treatment, differences in productivity losses, and differences in the consequences for carers. Whilst the PTO questions aim to control for some of these additional aspects, and the survey can attempt to find out whether that has been successful, an in-depth interview is required to fully explore respondent's motivation for their answers.

Questions which focus on the single issue of age may give an indication of the maximum strength of preference towards giving additional weight to improvements in health based on patient age, as the respondent's attention is focused on age. An additional criterion of patient life expectancy was tested in the pilot interviews for one of the quality of life PTO questions with the aim of being able to explore the relative strength of preference towards patient current age versus another relevant prioritisation criterion. However, during piloting it was found that some participants made the assumption that the life expectancy was a feature of the program treatment (despite our attempt at clear instructions to the contrary); hence, rather than representing an additional fairness criterion, this was interpreted as a difference in the health gain between groups. These questions were therefore not taken forward to the main study.

## 2.3 Person trade off survey design

**2.3.1 Perspective adopted by the respondent.**   Members of the public are simultaneously users or potential users of healthcare for themselves and their families, citizens, and payers in collectively funded healthcare systems. They could therefore respond from any one of these perspectives. A commonly adopted approach in studies eliciting preferences on healthcare prioritisation is to encourage responders to place themselves in the position of a social decision maker for a resource-constrained health system. This 'veil of ignorance' (Rawlsian) perspective has been proposed as the most appropriate option for social decision making [17]. In this study, respondents will be asked to adopt the position of a social decision maker. The survey asks respondents to imagine that they have been asked to help advise a healthcare decision maker (for example someone working in government).

**2.3.2 Framing of what respondents are encouraged to consider.**   When comparing health conditions and healthcare for different age patients, respondents may consider age-based differences in effectiveness of treatment, side-effects, treatment costs, impact on family members, the impact of health or treatment on productivity, and impact on education or social outcomes. As some of these issues can be internalized within standard economic evaluation methods the PTO question would ideally focus the respondent's consideration only on those aspects which tap into differences in the value of child versus adult health that cannot be captured in other ways, e.g. the fact that children are especially vulnerable or the fact that they have lived for fewer years than adults (had a shorter 'innings' so far).

Some PTO studies explain the change in patient group size via an imagined change in treatment cost. For example, in McHugh et al [34], respondents were asked a standard initial PTO question and then asked to imagine that the cost of their preferred of two treatment options changed, meaning fewer patients could be treated in their preferred group, while the other treatment option could still treat 100 patients. However, other studies adopting this approach have counterintuitively found a shift in preferences towards the populations that were more costly to treat [38], suggesting respondents incorporate other criteria into the choice when told the costs of treatment alter.

Our choice of study design aims to isolate the effects of prioritisation based on age of the patient from a) consideration of spillover effects on carers, b) productivity effects and c) assumptions about relative costs of treatments. We do this by informing respondents that they will be advising decision makers to choose between two treatment programs which are targeted at conditions that have a different age profile of patients which have the same costs, the same impact on the patients' carers in terms of their health and wellbeing, and the same impact on carer and patients' ability to work. This discussion occurs within the introductory video, and the assumptions are re-iterated at the start of each PTO question. A section of the text used in the video is shown in Box 2 (the full video can be found here): https://youtu.be/SX1bZRChza4.

**2.3.3 Type of health gain considered.**   Health gain can be framed as Quality Adjusted Life Years (QALYs), years of life or quality of life. The relative difference in value between child and adult gains may differ based on the type of gain being considered. Ultimately, much HTA in Australia and elsewhere relies upon Incremental Cost Effectiveness Ratios expressed in terms of cost per QALY, hence it would be useful for decision makers to know relative values framed as QALY weights. However, the use of QALYs within a question would require the respondents to understand this concept. This is likely to be an additional comprehension challenge and add to responder burden. If wishing to consider potential differences in QALYs gained through life extension and QALYs gained through quality of life improvements, then this distinction would need to be included in addition to the QALYs, i.e. a gain of 1 QALY

Box 2. Introduction to proposed survey (shown in the introductory video).

This survey is all about prioritizing healthcare. We would like you to imagine you have been asked to advise **a healthcare decision maker** (for example someone working in government) who has to make difficult choices about which treatments to fund. Sometimes those treatments are for patients of different ages.

Decision makers will want to consider the overall costs and benefits of new treatments. They can measure improvements in the health of patients – which could be extra years of life or improved quality of life. They can measure some other things that might differ like:

* The overall costs

* The impact on the lives of caregivers (who could be parents or spouses of the patients for example)

* The impact on the patients or their caregiver's ability to do their jobs and on their income

What they don't know is whether people in society think there is something else which makes providing better health to different age patients more or less important.

Is the same **health gain more important or more valuable for patients of different ages?**

In the survey we are going to ask you to choose between two treatment programs – imaginatively named A and B. The treatment programs will differ across the questions but the amount of health gain is always identical for the patients in both A and B.

We want you to imagine that the decision makers have already found out that there is **no difference overall** between the programs in terms of cost, impact of carers, or on people's income. Just for the sake of these questions, we want you to imagine that **the only difference is the age of the patients.**

We want you to tell us which program the decision makers should choose.

---

derived from a quality of life improvement. Likewise, if different types of quality of life improvements have different relative weights, this also needs to be considered. For simplicity, the PTO questions in this study will consider improvements in length of life (spent in full health) only and quality of life only.

**2.3.4 Number of years of life extension.** Previous PTO studies using additional years of life consider gains from 1 year [39] up to 40 years [26]. When wanting to understand the value of extending life, it is impossible to avoid the fact that this is only relevant when an individual is close to death and yet, ideally, we would prefer to avoid 'end of life' considerations which might incorporate other values which interact with age [40] (for example, adults might be perceived as needing time to get their affairs in order; young people may be perceived as needing time for parents to emotionally prepare for their death). This would suggest a preference for using a longer time-period within the PTO questions.

Some surveys have tried to frame the question through spending time in a coma or being bed-ridden [36], thereby valuing the time spent during a particular age but avoiding the considerations around death. Another option is to suggest that patients may be eligible for future funding [20], however, this raises an additional uncertainty in terms of what respondents actually consider relating to this unknown future outcome.

A particular challenge with years of life gain in our context is the changing age of the groups during the period of the years of life gain; a 10-year gain for a 5-year-old versus a 20-year-old, for example, moves towards trading between a 15-year-old versus a 30-year-old. Holding the patient age in years constant would require a period below 1 year (i.e. the patient was aged 10 at the point of the decision and the additional extension of their life is below 12 months so that they remain aged 10 throughout that period), which also has the benefit of being a marginal health gain as preferred in decision making (see above). These two considerations (avoiding treatment gains occurring at the end of life and avoiding treatment gains during which the patient groups get older) conflict with each other. Ease of interpretation for respondents is an additional consideration, which favours using whole years.

In this study, we will ask questions using two different life extensions (2 and 5 years) to test the difference this makes to preferences. These time lengths were chosen to limit the potential for responses to be heavily influenced by being close to death, whilst also limiting the potential variability in interpretation of the age of the groups and generating values for reasonably marginal health gains.

**2.3.5 Choice of quality of life domain.**   The quality of life loss which is averted by the hypothetical treatment in the PTO question could cover any aspect or domain of health-related quality of life or could be described in terms of a generic percentage loss in overall quality of life. There may be some dimensions of quality of life in which the relative value of improvements in that dimension for a child versus adolescent versus adult may differ e.g. the importance of being able to manage personal care.

The quality of life domains for our study PTO questions needed to be relevant across all ages that are to be included in the study (some aspects of quality of life may be unique to adulthood e.g. aspects of independent self-care). The domains also need to be easily described and understood. Given the potential for age-based prioritisation to differ across different types of health loss, the study includes mental health and two different types of physical health. Specifically, health states include:

i. 'Distress (low mood and anxiety)'. This is chosen as a common mental health problem. The choice of term 'distress' is chosen as this can be applied across all age groups including infants.

ii. 'Problems walking or moving about'. We chose to include the term 'moving about' in addition to 'problems walking' as alone this is not appropriate to describe health loss in a 1-month-old.

iii. 'Pain'. We chose to include pain as this is a common health problem experienced across the age range and is the dimension often judged by the public to be most important in health state valuation studies.

We acknowledge that there may be asymmetry in the value function by age when considering health gain versus health loss; however, exploring this is beyond the remit of this study.

**2.3.6 Without-treatment counterfactual and remainder of life course.**   PTO questions which involve an extension of life are also questions about delaying death. Typically, no benefit can be experienced by the groups beyond the end of treatment and the questions are framed such that death occurs immediately after the treatment period. As noted above, some studies

have tried to find a way of patients' life continuing after the life extension period by asking about patients avoiding being in a coma (during which they would gain no valuable experience from their life making it broadly equivalent to being dead). However, this raises additional issues and complexity e.g. introducing implications for parents/families, which may affect results but which would not arise in most treatments which extend life.

For quality of life improvements, death does not need to follow the period of health gain (compare Fig 2 with Fig 1). If the period of health gain is followed by additional years of life, there is a risk that respondents will think about differences in future potential health impacts by age and ignore the given information that the health gain between the two age groups is intended to be exactly the same. Respondents may, for example, prefer to treat younger patients if they think the treatment will avoid long term health implications. This raises two concerns; firstly, the comparison between the relative weights derived from quality of life gain and life extension gain is problematic if they are not comparing like for like. Secondly, the age-weights may be biased if respondents do not interpret the health gain to patient groups as the same thing.

These concerns could be addressed if the quality of life improvement was followed by death (see Fig 2). However, given the potential for the closeness of death to impact upon respondent preferences, if most health interventions to increase quality of life are not targeted at the end-of-life period then framing the treatment period as being followed by death may limit the generalizability of the age-weights. If death does not follow the enhanced quality of life period, then questions must be framed as either returning to full health (Fig 3) or remaining (or returning) to below full health level (Fig 4). The duration of the treatment quality of life gain could adopt any number of years, but a longer duration will change the age of the patient group.

Given the potential challenges of interpretation with other framings, the study PTO questions about treatments extending years of life will be followed by death. PTO questions about

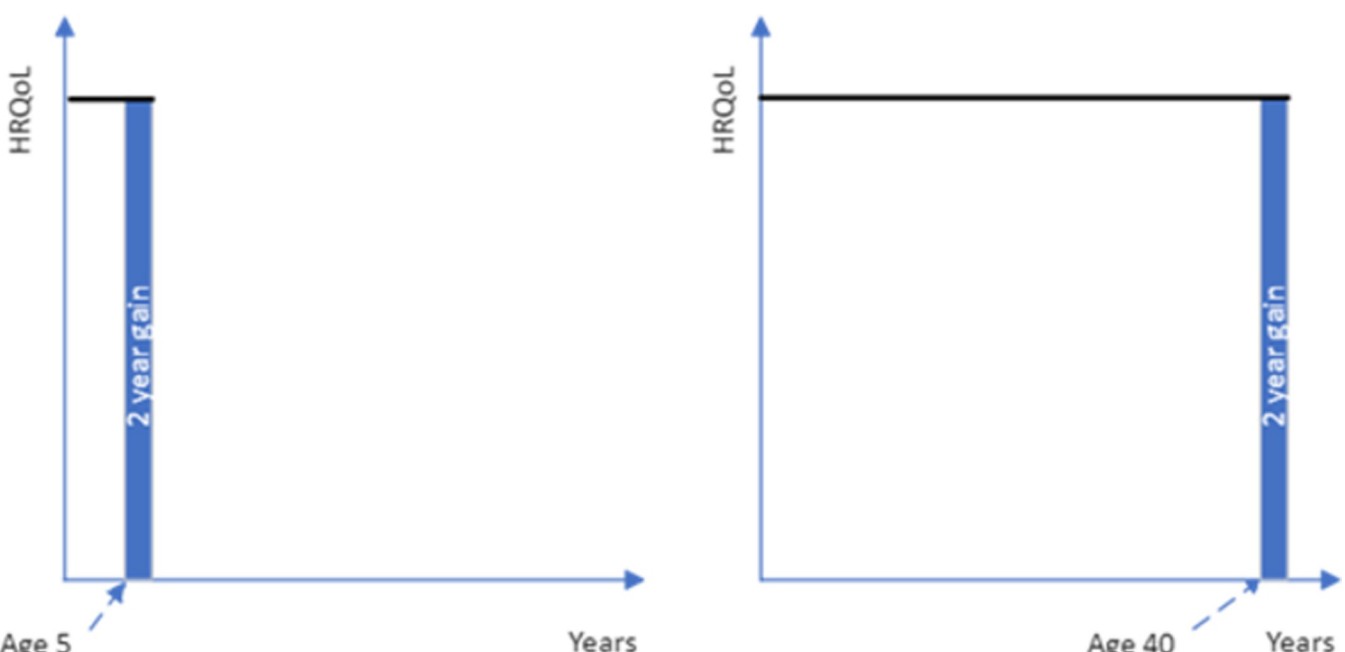

**Fig 1. A life extension for 2 years for a 5-year-old versus a 40-year-old after which time they will die or require additional funding (blue shading is the treatment gain being considered, without treatment patients would die at age 5 or 40, the black line is the health profile with treatment).**

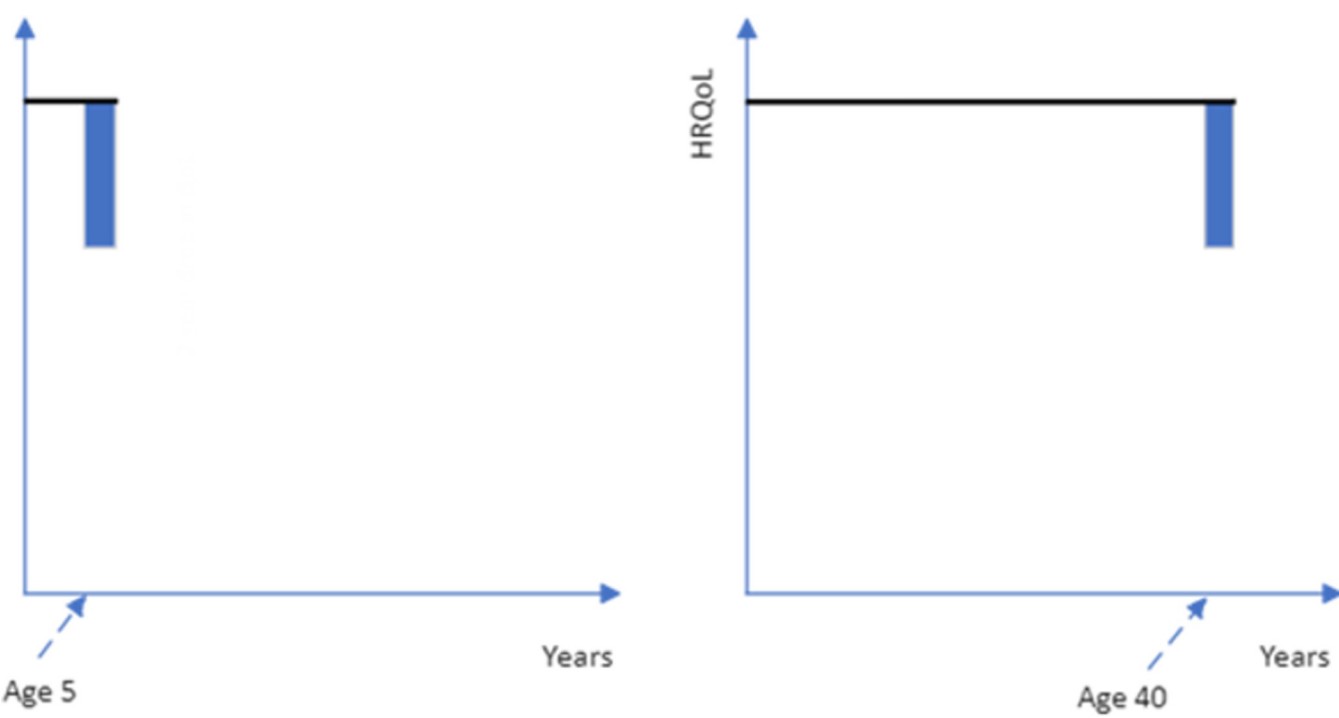

**Fig 2. Quality of life deficit for two years for a 5-year-old versus a 40-year-old after which time they die (blue shading is the treatment gain being considered, without treatment patients would incur a drop in their HRQoL, the black line is the health profile with treatment).**

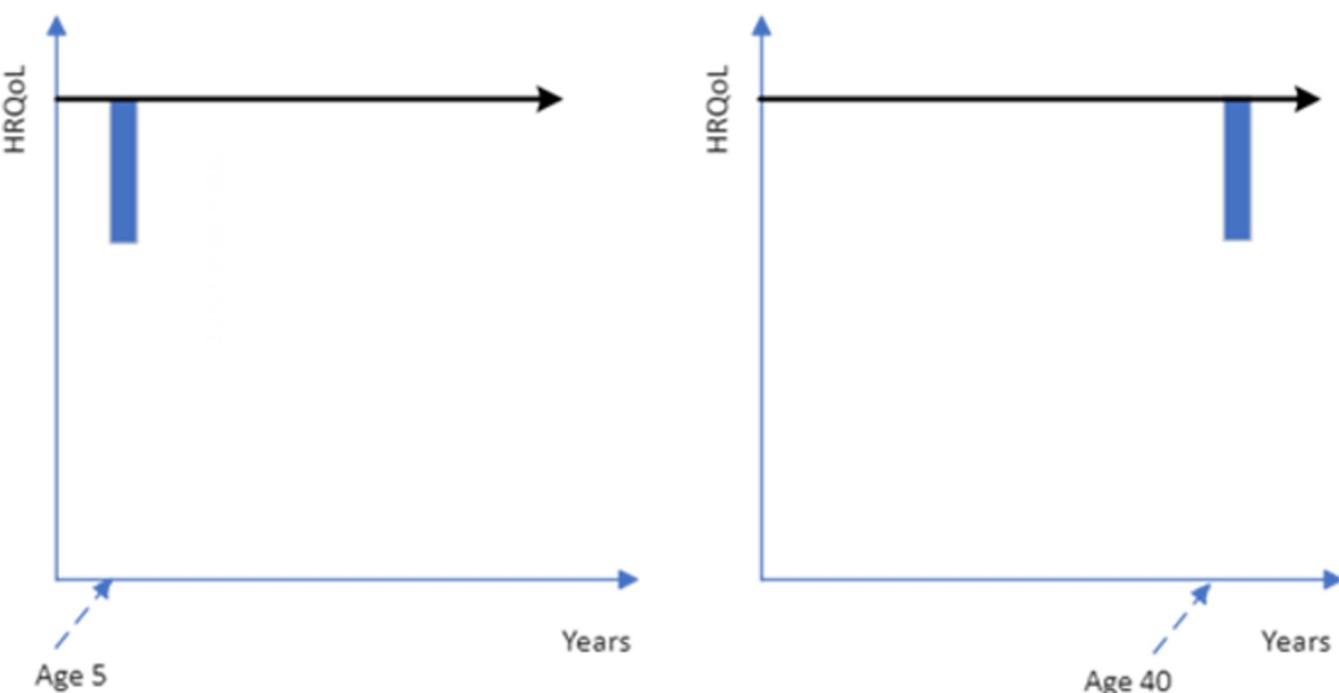

**Fig 3. Quality of life deficit for two years for a 5-year-old versus a 40-year-old after which time they return to full health for specified life expectancy (blue shading is the treatment gain being considered, without treatment patients would incur a short-term drop in their HRQoL, the black line is the health profile with treatment).**

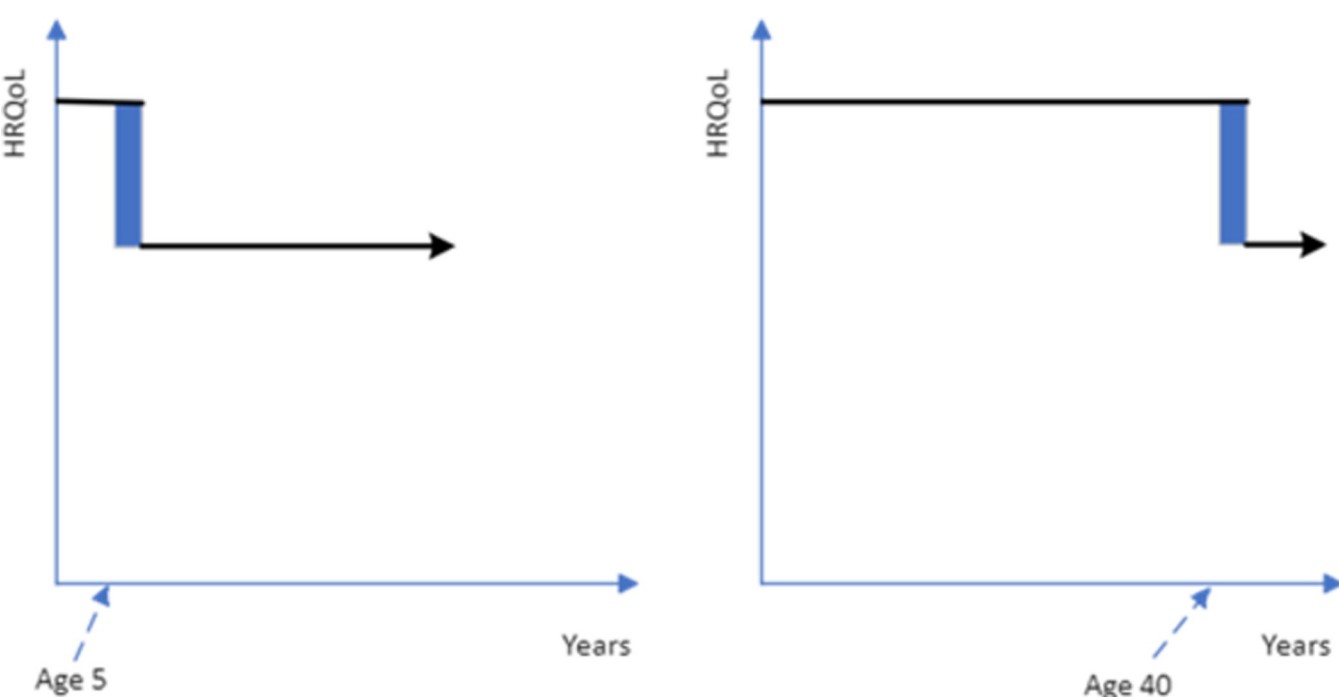

**Fig 4. Quality of life deficit for two years for a 5-year-old versus a 40-year-old after which time they return to the level of quality of life they would be at without treatment (blue shading is the treatment gain being considered, the black line is the health profile with treatment).**

treatments temporarily improving quality of life will be over a period of 2 years with the patients returning to full health after that period; for example, the health gain is described as a treatment which '*Prevents a 2-year illness with the symptom distress (low mood and anxiety) after which they would return to normal health with no long-term health consequence*'.

**2.3.7 Group ages.** In choosing the ages of children and adults for comparison within the PTO questions several factors were considered, including ease of interpretation, feasible sample size for each age comparison and existing gaps in the literature. The age category 'newborn' is often used in studies exploring social value, but this may be a slightly ambiguous category since a small baby a few months old could be described as newborn, as could a baby a few minutes old. A common childhood age used in the literature is 10 years old, therefore including this category would enable a direct comparison of findings. The recent systematic review [11] identified gaps in the evidence base in relation to very young children (<5 years) and young adults (18–24 years). It also identified a non-linear relationship between age and social value (lower for the very young children and adolescents compared with primary school age children) which suggests there would be a benefit to including many different young ages within the study.

The study adopts thirteen age categories (1 month, 2, 4, 6, 8, 10, 12, 14, 16, 18, 20, 22 and 24 years) enabling a reasonable number of respondents answering questions on each age category whilst including sufficient age categories to identify the hypothesised non-linearity of the value of health gain across different ages. The study also seeks to understand the relative preferences held towards health gains for patients as they transition from childhood to adulthood, hence three young adult age categories are also included.

To ensure a reasonable sample size for each age comparison (given the fixed study budget), only two adult ages were used in the main questions (with two other child/youth ages used in the chaining questions). We chose adult ages which would unambiguously be considered

'adulthood' but would not evoke issues around health and healthcare in older ages, hence pick up a potential preference to dis-favour very old adults compared to younger adults. In piloting, some respondents prioritised 40 year olds over children due to an assumption that they may have young children as dependents, therefore we complemented this age with an older age, 55 years, which is less likely to be associated with parents of small children.

The social value of health gain to young and middle-age adults versus old adults is outside the scope of this study.

**2.3.8 Choice of group size in terms of number of patients.** Common group sizes used in PTO studies are 10 or 100. A potential concern within PTO studies is that responders may consider the absolute difference between the number of people within each option of a PTO as well as ratio differences resulting in different ratios depending upon initial group size. The smaller absolute size may lead to greater risk of absolute differences impacting responses as the decision may feel more personal with fewer patients. Petrou et al., (2013) [9] tested the impact of a PTO with 1000 versus 100 patients and found this to have minimal impact upon the esti-mated weights.

PTO questions are at risk of start point bias [41]. An option commonly used to address and test for this is to randomly adopt different starting points. Unlike when using the method to compare different types of health gain (i.e. to estimate social utility values), when comparing different ages there is a policy default option that gains are of equal value therefore anchoring bias of equal value is a reasonable position.

This study adopts a starting point for the PTO questions of 100 patients in both groups. This is an easy number for respondents to work with and maintains more statistical anonymity than using 10 patients. The starting point bias will therefore be in favour of equity.

**2.3.9 Process for reaching equivalence.** In choosing the process for reaching equivalence, considerations include whether to allow the respondent to express equivalence in any of the questions, and the size of the groups to be shown to respondents.

It is common within the literature to require respondents to express a preference between patient groups in a PTO question (e.g. Baker et al., 2010 [28], Petrou et al, 2013 [9]). Forcing participants to decide which group should receive treatment has been seen as a means of ensuring respondent engagement. Damschroder and colleagues note that if a respondent states that the treatment options are equivalent this might mean that they hold them to have equal value, but it could also mean that they think *"the kinds of choices presented by the PTO should not ever be made. Alternatively, the subject may not understand the task or may not have taken the task seriously and simply took the quickest path through the survey."* [42](p 173). Nord et al noted, *"It would not be surprising if some avoided the issue simply by maintaining that all patients should be equally entitled to treatment, even if this is not possible in the real world. To the extent that this occurs, public responses to person-trade-off questions will not be of very great help in decision making about resource allocation"* [24](p 207).

However, holding a view of no preference between treating patient groups of the same size yet different ages is a legitimate preference. Given the potential impact of focusing effects, encouraging respondents to commit to a preference for one age group in the first question may impact upon responses to later questions and thus overall results. The current literature does not have sufficient comparable studies to explore this, hence the full impact of removing an option of equivalence when group sizes are equal is unknown. This PTO will incorporate a methodological test on the impact of allowing respondents to express indifference between age groups. Respondents will be randomised to either be given the opportunity to express no pref-erence or be forced to choose between the two options. Through the qualitative work and com-parisons to other parts of the survey we will try to discern whether respondents who opt for equivalence are:

i.  trying to find the quickest way out of the study

ii.  avoiding making a decision because it is unpleasant to do so

iii.  avoiding making a decision because it is too difficult and they feel unable to

iv.  expressing a genuine preference of equivalence

v.  failing to understand the question

To counter respondents learning that selecting the equivalence option is a quick way to progress through the survey, we will display an additional question when equivalence is chosen at the first question in which group sizes are both 100. This will always show group sizes of 100 versus 75 and therefore also acts as a potential consistency check to help identify genuine preferences.

As part of the process of moving to equivalence, a decision is required on which group size to vary; always the same patient group (e.g. always children), the preferred group, or the least preferred group. If respondents hold different views on which group represents the most valuable health gain, then always holding the same patient group at 100 patients will result in some equivalence values being above 100 and some below 100. When the patient group for the preferred option is reduced in size, this is naturally bounded by the size of the group, i.e. it cannot go below one patient. However, when the patient size of the least preferred group is allowed to vary, this will increase and have no natural upper limit.

In this study, the questions following the initial choice will depend upon respondents' answers, with the least preferred group from the initial choice always remaining at 100 patients. For example, if participants prefer Program A, then this patient group size will be reduced in size. Thus, we will be able to avoid very extreme values as the lowest ratio is 1:100.

Some equivalence studies ask respondents to directly state the equivalence values; however, where this has been piloted in other studies, it has generally been perceived as problematic and difficult for responders (e.g. Al-Janabi et al, 2022 [43]). An alternative approach (drawing upon best practice for willingness to pay studies) is to present a single PTO question with a range of different group sizes (e.g. Eisenberg et al, 2011 [44]), although this limits the data that is provided. A more common approach is to iteratively move towards equivalence through a number of follow-on questions. The number of iterative questions used in PTO questions is typically around 4 (e.g. Reckers-Droog 2019 [37], Al-Janabi et al, 2022 [43]) to 6 (e.g. Petrou et al, 2013 [9]), although some studies, particularly those with interviewers, use as many as it takes to reach equivalence. Equivalence can be reached by the 'ping-pong' method (oscillating up and down in terms of relative group size), a gradual increase or decrease of one group size ('titration'), or bisection which involves halving one group size initially, and for follow-on questions depending upon the previous response, either halving the group size again or using the midpoint between the two group sizes from the previous question or between the preferred group size from the previous two questions.

The 'ping-pong' search approach has an advantage of minimising the starting point bias [24]; however, in this case the starting point of 100 patients in each group is a defensible default position; therefore, starting point bias is less of a concern. The 'ping-pong' method can be confusing for respondents and has been found to take longer and have higher standard deviation than titration in standard gamble health valuation studies [45].

A bisection approach was chosen for this study with three or four follow-on questions, with the number of patients in the preferred group initially halved (to 50). Follow-on questions iterate towards the indifference point by roughly halving the remaining range of uncertainty but using numbers which are easy for the respondent to process (e.g. rounded to the nearest ten or

five patients). The number of patients within each question is shown in the flow chart (Fig 5) below. Our use of four (or for some choices five) iterative questions seeks to achieve a balance between the likely number of subsequent questions required prior to reaching equivalence, the survey duration, ease of comprehension and avoiding boredom for responders.

**2.3.10 Supporting the comprehension of respondents.** Given that the purpose of PTO studies is to inform healthcare policy, it is important that respondents understand the questions they are being asked and the implications arising from their answers. One option to support respondent understanding could be to show the aggregate health gain comparison implicit in the respondents' decision and ask for confirmation–thereby aiming to make the lower health gain in the preferred group implicit in the trade off choice more salient. For example–if the respondent opts for 50 young people over 100 older people and the health gain is 2 years life extension they could be asked "That would mean that you would prefer treatments giving 100 years of additional life to patients aged X than an identically costing treatment which gave 200 years of additional life to patients aged Y–are you happy with that choice?" This could be included in the initial questions to both increase respondent understanding and to improve confidence in the respondents' choices. However, there are some problems with this:

i.  Quality of life improvements over time can only be aggregated to QALY gain if they are expressed as a percentage of full quality of life (e.g. on a VAS scale)

ii.  If an opportunity for respondents to change their minds is included in some questions but not all, then a direct comparison between questions with and without this option is no longer possible.

iii.  It introduces additional complexity and time for the respondent.

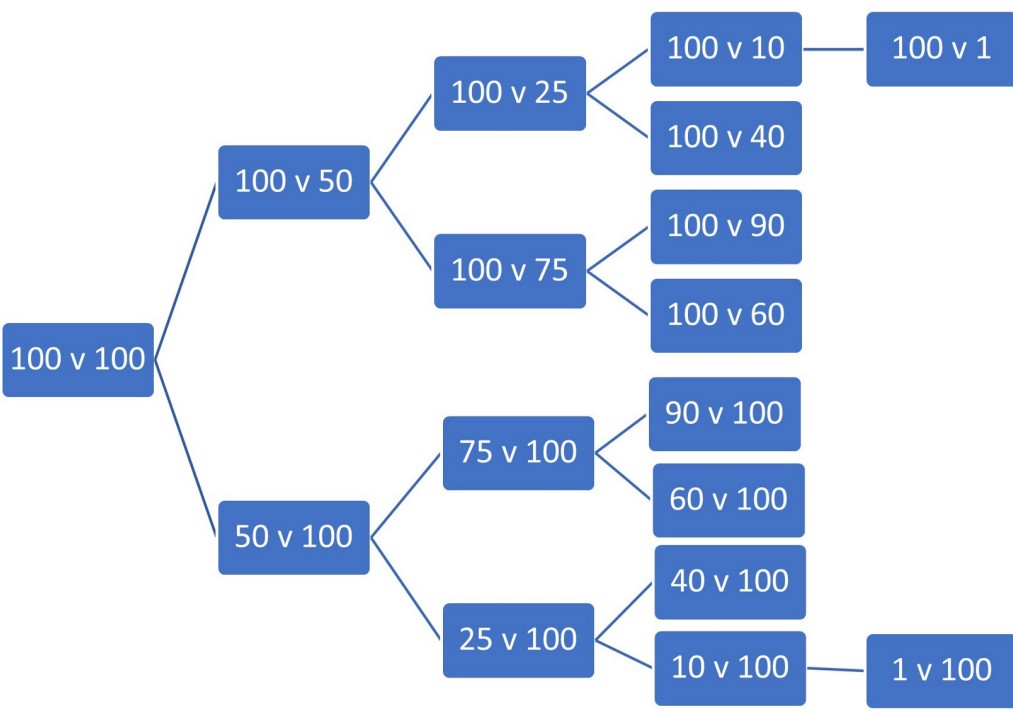

**Fig 5. PTO iterations of the question flow showing number of patients in Program A vs Program B.**

iv. A feedback check which states the years of life lost arising from the choice frames the choice within a utilitarian framing. This might then influence respondent's subsequent preferences.

Several previous PTO studies have used a ranking exercise in part as a warmup (e.g. Petrou et al, 2013 [9]) and in part as an internal consistency comparison. Whilst ranking data may provide a useful comparison, it does not provide a warmup of the same style as subsequent questions within a PTO survey and can take a considerable time.

Several steps will be taken in this study to support good understanding of the survey by responders. This includes extensive testing and piloting of the survey, including a mandatory introductory video to help respondents understand the purpose and how to answer questions and including a 'back' button to enable responders to change previous answers where they may have made an accidental mistake.

**2.3.11 Testing data quality and the stability of preferences.** Internal consistency of responses can be assessed by tests of cardinal transitivity [46], which is also referred to as ratio consistency, multiplicative transitivity or triad consistency. An individual's PTO equivalence value for ages 10 versus 20 and 20 versus 30, for example, should accurately predict their PTO equivalence value for age 10 versus age 30. To test this at an individual level requires individuals to be asked at least three questions which 'chain' together.

Internal consistency may be judged by comparing PTO answers against other methods for eliciting aged-based preferences such as opinion questions and ranking. However, past evidence suggests fairly systematic differences across research methods with ranking more often being linearly related to age, and opinion surveys responses more likely to reflect a preference for equal treatment [11]. Therefore, these indirect comparisons cannot say much about data quality.

This survey will include an attention check question, consistency comparisons to screening data based on personal demographic data and a minimum time to complete, all of which aim to exclude responders who are not engaged with the survey. We will also consider consistency comparisons to attitudinal questions, but these will not be treated as strict quality checks since there may be legitimate reasons for these differences.

A ratio test will be undertaken for one of the years of life extension questions by chaining PTO values. Each respondent will answer the main PTO questions for just one age. For the question relating to life extension of 2 years they will answer two additional chaining questions i) their given age compared to an age which is either higher or lower by 10 years, as shown in Table 1A, and ii) this age compared to aged (40 or 55), as shown in Table 1B. The expected PTO value will be calculated for each individual based on the two PTO equivalence values (Age X vs (40 or 55)) x (Age Y vs (40 or 55)) and this will be compared to the actual

**Table 1.** a. A given age compared to an age either higher or lower by 10 years. b. Age Y (values from Table 1a) compared to age (40 or 55).

| Age | Ages used in the PTO for life extension of 2 years | |
|---|---|---|
| X | Randomly chosen from one of 13 ages (1 month to 24) | |
| Y | Age X plus 10 years if X $<= 12$ OR Age X minus 10 years if X $>= 14$ | |
| | Expected value | Actual value |
| If X is the youngest age | (Age X vs Y) x (Age Y vs (40 or 55)) | X vs (40 or 55) |
| If Y is the youngest age | (Age Y to X) x (Age X to (40 or 55)) | Y vs (40 or 55) |

equivalence value (Youngest from X or Y vs (40 or 55)). For example, if the respondent is given age 8 for their PTO questions, they will be asked to complete a PTO comparing patients of ages 8 vs (40 or 55). They will also be asked to compare ages 8 vs 18 and ages 18 vs (40 or 55). The product of the last two responses (the expected value) will be compared to the ages 8 vs (40 or 55) response (the actual value).

**2.3.12 Order effects and randomisation of question order.** Order effects may arise through learning effects which result in respondents giving more accurate preferences in later questions and through respondent fatigue which may result in less attention being given to later questions. The order of questions may also generate framing effects leading respondents to focus on certain aspects of the choice–for example asking a question about children leading short lives first may lead people into considering length of life even during questions on quality of life. Earlier questions may draw respondents' attention to differences between temporary versus permanent health gains [47]. Ubel et al., (2001, 2002) [48,49] identified strong order effects in PTO questions both from numerical anchoring from the group sizes in earlier questions and from questions which make people think about equity considerations and avoiding discrimination. To avoid ordering effects, this study will randomise the order of the PTO question, although the survey will avoid switching question types (e.g. between life extension and quality of life improvement) so as not to add to respondent burden. Participants will be randomised to receive either the life extension questions first (4 questions shown in random order) or the quality of life enhancing questions (3 questions shown in random order) first. The attitudinal and opinion survey questions will follow the PTO to try and avoid further focusing effects. We will test the impact of participants seeing life extension or quality of life enhancing questions first.

**2.3.13 Use of icons and pictograms.** The use of images, icons and pictograms within preference elicitation studies is common. Their use is seen as supporting respondent understanding and speed of interpreting the scenarios presented. It is also a means of improving the look and feel of the survey to a more pleasurable respondent experience.

Icons could be used to represent the age of respondents (Fig 6); however, it is difficult to find a suitable icon which clearly differentiates across all the ages included in the study design from 1 month to 24 years and aged 40 and 55 required for this study. Icons can help represent the number of patients in the group (Fig 6); however, the space taken up by these images is considerable–which may undermine the ability of respondents to complete the survey using a

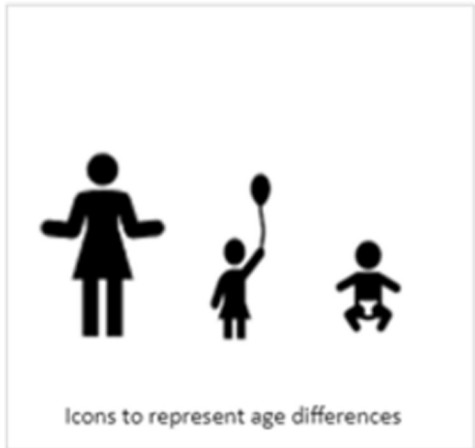 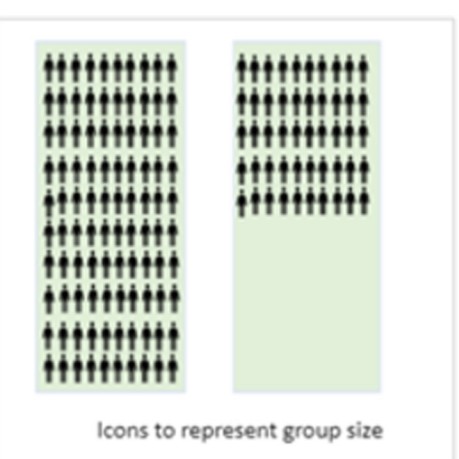 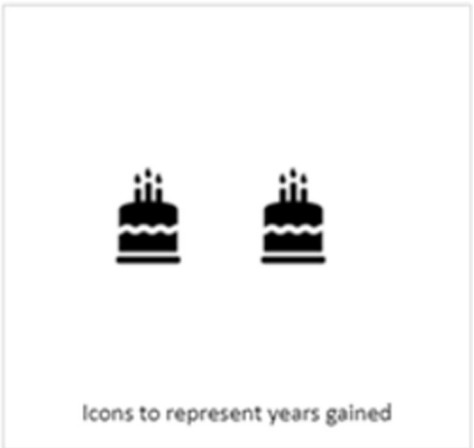

Icons to represent age differences    Icons to represent group size    Icons to represent years gained

**Fig 6. Potential icons to support presentation of PTO choices.**

mobile phone–hence limiting likely response particularly amongst younger responders. The number of additional years of life could be presented visually ([Fig 6]), but improvements in quality of life are harder to unambiguously present visually.

No icons were adopted in this study in part due to space considerations and in part because icons could not clearly distinguish all ages, life expectancies and health gains. Using icons for some attributes but not others may focus attention on the attribute with the icon.

**2.3.14 Choice of socio-demographic questions.** Socio-demographic questions serve four purposes: 1) describing the sample of responders, 2) assessing the extent to which the final sample is nationally representative, 3) testing for differences in comprehension, data quality and preferences across socio-demographic characteristics, and 4) providing an opportunity for a consistency check to test the concentration of responders where questions are asked more than once (e.g. age and age-group).

There are some demographic questions which, based on previous work, we hypothesise to influence the relative value placed on child health; these include age, gender, and parenthood status (which may include having young children, older children, and grandchildren). Familiarity with serious illness, either for oneself or one's children, may also impact upon preferences. These variables are therefore included.

Some additional demographic questions are included to enable judgement of the representativeness of the final sample relative to the Australian population. This includes location (State/Territory), education, and ethnicity. In addition, education status may impact upon comprehension of the questions. Education is also likely to act as a reasonable proxy for income status therefore will also be useful for assessing the representativeness of the sample.

**2.3.15 Choice of feedback questions on comprehension.** Feedback questions asking about respondent understanding can provide some indication of the data quality and consequently the value of the final data for supporting health care decision making. Responses may help identify very low-quality data for which there may be justification for exclusion. This data may also be valuable to evaluate the quality of the research and contribute towards future improvements in research methods. Considerations in the choice of feedback question(s) also include the willingness of respondents to truthfully reveal their responses, which may be unlikely if respondents think certain responses may be linked to receiving the completion reward or future research opportunities. Two questions will be asked in this study on respondent comprehension:

i. How well did you understand the questions which asked you to pick between Program A and Program B? (I found them easy to understand / I found them reasonably easy to understand / I found them a bit hard to understand / I found them really hard to understand)

ii. Would you be happy for your responses to be combined with other people's to be used in health care decision making? (*yes / not sure / no*)

These responses will be used to describe the level of self-perceived comprehension of responders. Subject to a degree of caution (as these responses cannot confirm whether respondents actually interpreted questions in the anticipated manner), we will consider the robustness of the results to the exclusion of respondents reporting low levels of comprehension and unwillingness to be used in decision making.

**2.3.16 Choice of feedback questions on reasoning.** Feedback questions on the respondent's reasoning are included to help understand something about the drivers behind peoples PTO responses and to assess whether they answered in the expected manner (e.g. ignored costs, and health effects outside of the time period in question). One potential problem may be

that different reasons are behind the different types of questions (e.g. quality of life or length of life) but asking after each question would be time consuming for respondents.

After all the PTO questions have been asked, respondents will be asked a question about the factors that influenced their answers (see Fig 7).

**2.3.17 Choice of attitudinal questions.** Attitudinal questions are valuable for decision makers in their own right; they also enable comparison to other numerical style trade-off questions within surveys. Differences between these question types can both highlight problematic responders and shed light on respondents' thinking and principles. Attitudinal questions should be easy to read and interpret (hence a preference for short statements). There may also be some advantages in replicating previously asked questions to enable a direct comparison to earlier work. The attitudinal questions chosen draw upon those included in Nord (1995) [17], Richardson (2017) [50] and Rowen (2016) [32]. These are shown in Box 3.

---

**Box 3. Questions included to explore respondent attitudes towards health care prioritisation.**

Which of these statements best reflects your views about prioritising different types of health care?

|  | Children should have some priority over adults. | Adults should have some priority over children. | People should have the same priority regardless of age |
|---|---|---|---|
| For medical care that improves quality of life temporarily (with no long-term effects) |  |  |  |
| For medical care that extends life by a few years |  |  |  |

If the Australian governments were willing to pay more for a treatment for children compared to adults which gave the identical health gain–what would you think?
○ This is fair because they are children
○ I'm not sure
○ This would be unfair

---

Which one of these statements best reflects your views about Medicare priorities?
○ Medicare should give priority to treating patients who will die young.
○ Medicare should give priority to treating patients who will get the largest amount of health benefit from treatment.
○ Medicare should give the same priority to treating all patients. Amount of health benefit and whether patients have had a short life is not relevant.
○ Medicare should base priority on a combination of treating patients who will get the largest amount of benefit and treating those who will die young.
○ Unsure
○ None of the above describes my views about Medicare prioritisation.

---

**2.3.18 Mode of administration.** Considerations regarding mode of administration include cost (and by implication sample size), data quality, and any potential impacts mode may have on PTO values. A previous comparison of face-to-face and online methods for PTOs found no statistically significant differences in PTO equivalence values, odds for giving extreme values, or measures of consistency when comparing face-to-face interviews with a web-based survey [42]. To maximise sample size the survey will be conducted online.

**Fig 7. Questions included to explore respondent reasoning.**

**2.3.19 Summary of the survey structure.** The survey flow is shown in Fig 8. The structure of the survey and the randomisation used (i.e. whether the respondent is shown an equivalence option or not, whether the respondent is presented the youngest age on the left or right of the screen, and which age comparison they see) is shown in Fig 9. A summary of attributes selected for the survey can be found in S3 Table.

## 2.4 Qualitative and deliberative component

**2.4.1 Qualitative 'think aloud' interview alongside PTO; aims and outline.** The interviews will complement the quantitative PTO survey and support the main aims of the study shown in Box 1. Specifically, they will seek to:

i. understand how respondents approach the PTO questions, for example, which information do they focus on

ii. explore whether they think the PTO questions can identify the relative weight they would give to improving child versus adult health

iii. understand respondents' reasons behind their preferences

iv. understand any seeming inconsistencies between attitudinal questions and PTO responses

v. understand how strongly views are held through subjecting respondents' opinions and responses to alternative views and disagreement

vi. help interpret online survey findings.

To gain most value from the qualitative work, some interviews will occur prior to the main online data collection (hence can support the pilot and be a final opportunity to improve the survey design) and some will occur after the initial data analysis (hence can explore any topics arising from the main online study). The one-to-one interviews will explore how respondents interpret and respond to the PTO questions through encouraging respondents to talk about the questions ('thinking aloud') whilst completing the survey. The interviewer will raise the

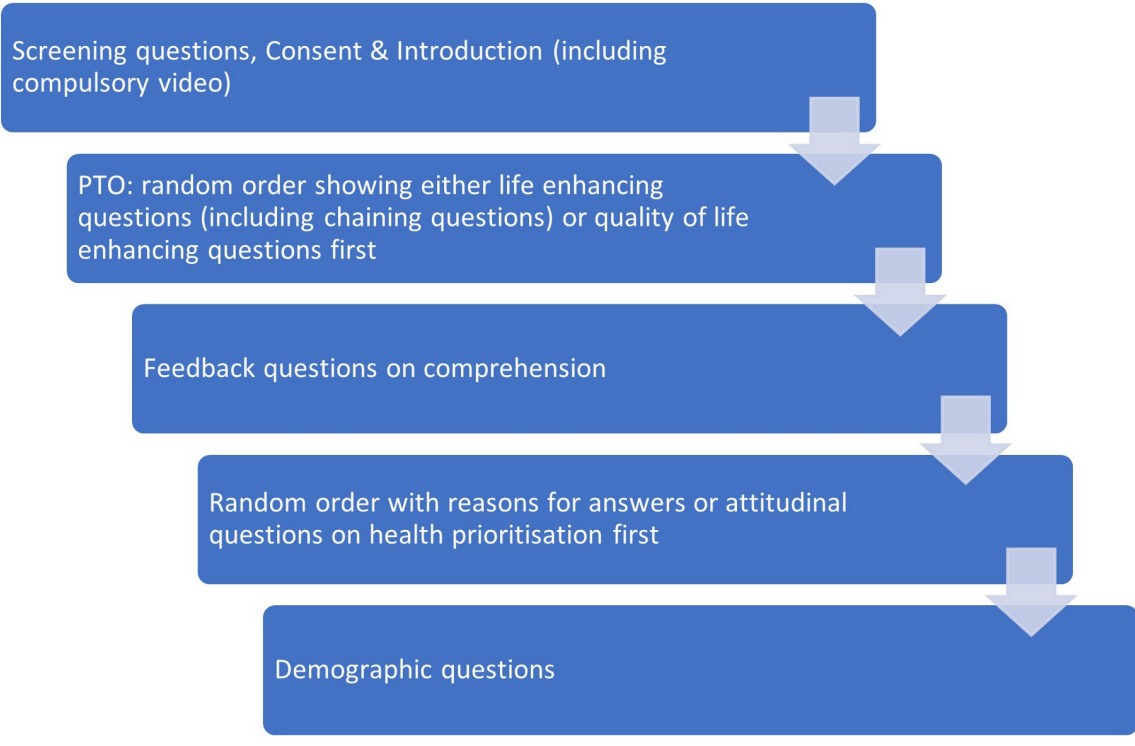

**Fig 8. Survey flow.**

fact that other people hold different or opposing views to explore the reaction to this indirect disagreement. The interviewer will highlight any potential inconsistencies within their responses which may shed light on any inconsistencies between attitudinal questions and PTO responses. The one-to-one interview prompts can be found in the S2 File. The interview protocol and risk management strategy can be found in the S3 File.

**2.4.2 Focus group aims and outline.** The focus groups will complement the quantitative PTO survey and support the main aims of the study. There is no clear normative justification for any particular method of aggregating preferences [51] and the focus groups will be a vehicle for exploring how respondents feel about aggregating potentially diverse preferences and exploring consensus reaching under situations of diverse views [52].

Specifically, they will address the following questions:

i. How do respondents feel about the use of PTO surveys to inform health care decision makers?

ii. How do they interpret the findings from the study?

iii. Do they think that something important is missing from the findings?

iv. How do they understand any inconsistencies between attitudinal questions and PTO responses?

v. How do they reconcile different opinions?

vi. Whose responses do they think matter most?

vii. Who do they think should make choices about use of age-weights?

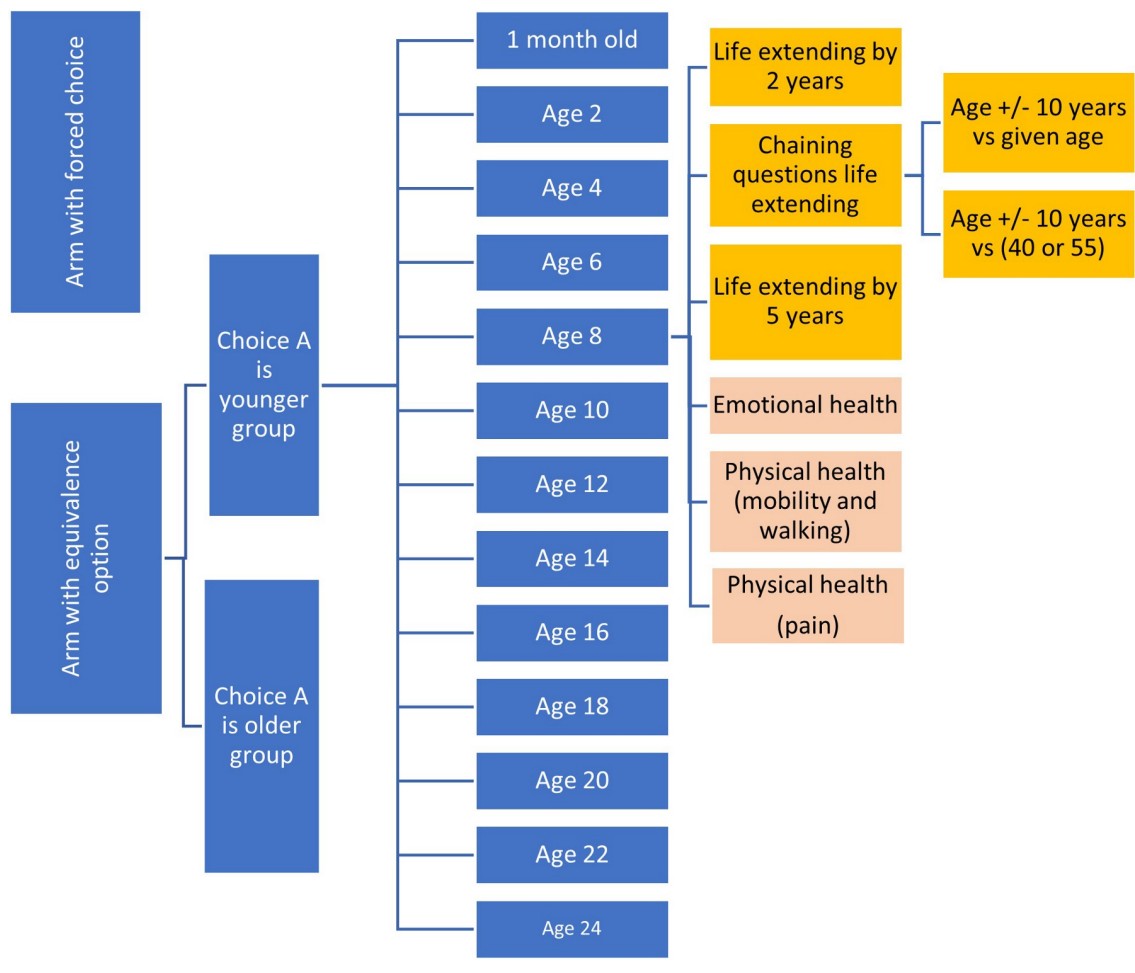

**Fig 9. Survey structure.**

The focus groups will take about 90 minutes and will be recorded. The focus group prompts can be found in the S4 File.

## 2.5 Sample size

**2.5.1 Sample size considerations and justification.** Sample size considerations include the precision of estimates and practical considerations in relation to running the study.

The PTO technique "*needs to be applied in fairly large groups of subjects to keep random measurement error at an acceptable level*" [17](p 207); however, the ratio data does not lend itself to specific sample size calculations. Based on existing literature, it was judged that about 50 responses was required for PTO each question. To enable exploration of subgroups the sample would need to cover a broad age range of respondents and parents with a child with and without a health condition, parents with younger and older children, and non-parents.

Informal feedback from online survey respondents suggests motivation during the self-directed online surveys drops when surveys exceed about 15–20 minutes.

**2.5.2 Sample size plan for quantitative PTO survey.** If each person responds to questions on one age group only (other than the chaining questions), given that there are 13 childhood/youth age groups (even numbers 1 month to 24 years) and two older age groups (40 and

55) to achieve 50 responders per question this would be a minimum of 1300 surveys. However, the sample size is greater to enable a comparison between the forced (in which there is no 'no preference' option) and unforced preference. The target 'online only' sample is n = 2000, of which 1000 will respond based on the PTO questions with only two options (forced preference) and 1000 will see the equivalence option (unforced preference). With successful recruitment, this should generate n = 38 forced and n = 38 unforced responses to each PTO age comparison.

**2.5.3 Sample size plan for 1 to 1 'think aloud' interviews and focus groups.** We anticipate saturation to be reached at a sample of 40 one-to-one interviews which is fairly large for qualitative research [53]. This should enable sub-samples to be considered including i) parents with young (< = 10 years) children, ii) parents with older children (>10 years), iii) young people (participants aged 16–24), and iv) adults without children. Given the tight focus of the interviews on participant views on one aspect of health care prioritisation only i.e. age, the assumption of saturation at this sample size seems reasonable.

Saturation here will adopt the approach of Guest et al (2020) [54] and "refers to the point during data analysis at which incoming data points (interviews) produce little or no new useful information relative to the study objectives" (p.5). A minimum (base size) of five interviews will be conducted for each sub-group, with a consideration after each additional two interviews (run size) as to whether new information or themes have been gained.

We will hold about four focus groups, each containing a maximum of 5 respondents to encourage in-depth discussion and participation. Our Consumer Advisory Group recommended very careful management of the participant experience given the sensitive nature of the topic and the inclusion of parents with sick children. In response to this concern, focus groups will be conducted with researchers available to provide the option of participants moving to a one-to-one interview should they prefer. Limiting the number of participants per group risks low attendance. However, as participants will have made arrangements to attend the session, the discussion will go ahead with any participants who attend. Low attendance may change the nature of the discussion and will be reflected upon during the data analysis.

## 2.6 Recruitment

**2.6.1 Recruitment considerations.** We aim to have a sample of responders who can represent the voice of the public. It is therefore important to consider the characteristics that might lead to differences in viewpoint. Based on existing literature, age, gender and parent status [11] have all been found to impact the relative value of child versus adult health gain. Quota sampling representative to the Australian population in terms of age and gender is straightforward. However, quota sampling based on parent status is less so. The 2021 Population and Housing Census (ABS) data records the percentage of households which have children aged 15 and under, and over 15 (including adults); however, a number of additional assumptions would be required to use this breakdown to set a quota for the percentage of each age/sex group with children.

It is also useful to reflect upon who would be most impacted by any subsequent recommendations from this research and whether the views of those likely impacted are included in the research. In this case, all age groups are potentially impacted (e.g. additional weight towards child health gains results in relatively less weight for adult health gains), suggesting coverage across age groups should be as representative to the population as possible.

Another important consideration is the inclusion of children's preferences. If all respondents reported a preference towards health gain to their own age group over other age groups, then it would be a concern if children were underrepresented within the sample. However, the

complexity of PTO questions creates a constraint in terms of respondent age, as does the potential for distress caused by thinking about poor health and limited life expectancy–particularly of children. At 16 years old, adolescents are considered sufficiently mature cognitively and emotionally to cope well with the questions; this was confirmed during the piloting phase.

Recruitment options include commercial research survey companies with their own (or access to other) panels, recruitment based on previous research panels, request letters sent by post, via organizations (health care, business), advertising via social media, or directly approaching people in the street or in their communities. Recruiting through commercial panels is fast and cheap but steps may be required to ensure data quality. Additionally, some groups may not be well represented on research panels.

**2.6.2 Recruitment plan for quantitative survey completed without an interviewer present.** Recruitment will use the commercial research company Online Research Unit (ORU) for the online survey recruitment. The ORU panel incentives are delivered to members only by post to a physical address; this has numerous data quality benefits and enhanced validation of respondents. Additionally, ORU panels are primarily recruited offline using post, phone and print. This should lead to improved representation compared to other panels that rely on predominantly online recruitment methods. Members of the ORU panel aged 16 and above will be invited to complete the survey. Drawing upon 2021 Census data, the sampling quotas will adopt broadly equal male and female recruitment in each of seven age-gender groups, with a slight over-sampling opportunity of about 5% in each of the age-gender groups to avoid excessive rejection of respondents at screening. Additionally, a quota will be applied based on attainment of a degree to ensure a balance of education levels which can also proxy for other socio-economic indicators. Loose geographical quotas will also be applied to give a reasonable state/territory mix. Respondents will complete a screening questionnaire and will only be able to enter the survey if their age/gender, education and location quotas have not yet been reached (see S1 Table for quotas). As our main concern is the ability to explore heterogeneity of preferences between parents and non-parents, it was not considered necessary to quota sample based on parent status. The data collection will be paused at a mid-point to explore the sample in relation to parenthood status and consider whether additional quotas are required for the remaining data collection.

**2.6.3 Recruitment plan for one-to-one interviews and focus group.** The qualitative work will purposefully target parents and non-parents, different age groups including young people (> = 16 years), and parents of children with a health problem. Recruitment will be via two mechanisms. The first sample will be recruited through a commercial company, CRNRSTONE (https://crnrstone.com.au/). This sample will be biased towards adolescents and adults who do not have children (< = aged 24), and adults who do not live with children, and will cover a mix of gender and ages. The second sample will recruited by following up on participants from a related earlier study (also funded from the QUOKKA MRFF project, referred to as the P-MIC study [55] who have consented to be contacted for future research (n = 984) (https://www.quokkaresearchprogram.org/research-1). This sample will only be parents or carers of children under 18 years old. Recruitment will continue until qualitative saturation has been achieved.

For the qualitative component, interviews will be arranged either by research staff or by the commercial recruitment agency. A reminder about the interview will be sent the day before.

## 2.7 Data analysis plan

**2.7.1 Description of the sample.**   We will report descriptive statistics of the sample characteristics for respondents who completed the interview for each arm. Sample characteristics will be compared to the 2021 Australian census data.

**2.7.2 Survey data quality assessment.**   Respondents who fail inclusion criteria based on quality (set out in 2.10.1) will be replaced by the survey company. We will report the numbers of responders failing these quality tests, but they will not be included in the main dataset.

The following potential quality indicators will be reported:

i. Respondents with potentially 'rapid completion' will be reported and dropped in sensitivity analysis.

ii. The percentage of incomplete surveys (note that little detail will be available on respondents who do not complete as many of the demographic questions fall at the end of the survey).

iii. Respondents in which the attitudinal questions are potentially inconsistent with responses to the PTO question.

iv. Respondents in which questions on comprehension suggest low understanding (e.g. How well did you understand the questions which asked you to pick between Program A and Program B?). Dropping these respondents will be tested in sensitivity analysis.

v. Respondents who report equivalence within the initial question (100 vs 100 group size) but for the next question in which they are offered 100 vs 75 report either equivalence or that they prefer the 75 group size will be considered in the context of their other responses. This apparent inconsistency may arise due to a desire to move rapidly through the survey, or it may reflect a view that one group of patients should not receive priority treatment over another regardless of age or number of patients in the program.

**2.7.3 PTO data analysis.**   Four iterative questions are asked for each PTO question which move towards an equivalence value, except where they report preferring 10 of one group to 100 of the other in which case, they see an additional question. In the arm where equivalence is offered as an option respondents may not reach equivalence by the last follow-on question. In these cases, and where equivalence is not offered as a choice, the respondent's equivalence patient group size can be inferred to fall within between their final and previous question group size. For these cases mid-point will be used (see Table 2), an approach adopted in other PTO work e.g. Al-Janabi et al, 2022 [43].

We will also infer a mid-point between 0 and the lowest possible number of patients that can be chosen for those responders who consistently prefer one age group over another (i.e. the lowest possible number of patients to choose is 1 rather than 100). This assumption will be discussed in the pilot and qualitative interviews should any respondents reach these values. Following McHugh et al (2018) [34], we will classify preferences in which less than 1 patient in one group is preferred to 100 patients in the other group as 'extreme preferences' and report the frequency of these.

Given that the arithmetic mean of individual ratios suffers from an asymmetric property (as can be seen by looking at the column 'Weight for Age 2 in Table 2), a simple mean of PTO responses is not possible. It is common practice (e.g. Petrou et al, 2013 [9], Pinto-Prades et al., 2014 [40], Baker et al, 2010 [28]) to show aggregated results as both a ratio of means (ROM) and median of the individual ratios (MOIR):

**Table 2. Assumption of equivalence group size based on final PTO follow up questions.**

| Final choice (Age 1 vs Age 2) | Response | Equivalence group size of preferred group | Equivalence group size assumed (adopting mid-point) | Weight for Age 2 |
|---|---|---|---|---|
| 100 vs 1 | 1 | <1 (undefined) | 0.5 | 100/0.5 = 200 |
| 100 vs 1 | Equivalence | 1 | 1 | 100/1 = 100 |
| 100 vs 1 | 100 | <10 & >1 | 5.5 | 100/5.5 = 18.18 |
| 100 vs 10 | 100 | >10 & <25 | 17.5 | 100/17.5 = 5.71 |
| 100 vs 10 | Equivalence | 10 | 10 | 100/10 = 10 |
| 100 vs 25 | Equivalence | 25 | 25 | 100/25 = 4 |
| 100 vs 40 | 40 | <40 & >25 | 32.5 | 100/32.5 = 3.07 |
| 100 vs 40 | 100 | >40 & <50 | 45 | 100/45 = 2.22 |
| 100 vs 40 | Equivalence | 40 | 40 | 100/40 = 2.5 |
| 100 vs 50 | Equivalence | 50 | 50 | 100/50 = 2 |
| 100 vs 60 | 60 | <60 & >50 | 55 | 100/55 = 1.82 |
| 100 vs 60 | 100 | >60 & <75 | 67.5 | 100/67.5 = 1.48 |
| 100 vs 60 | Equivalence | 60 | 60 | 100/60 = 1.67 |
| 100 vs 90 | 90 | <90 & >75 | 82.5 | 100/82.5 = 1.21 |
| 100 vs 90[a] | 100 | >90 & <100 | 95 | 100/95 = 1.05 |
| 100 vs 90 | Equivalence | 90 | 90 | 100/90 = 1.11 |
| 100 vs 100 | Equivalence | 100 | 100 | 100/100 = 1 |
| 100 vs 90[b] | 100 | >90 | 100 | 100/100 = 1 |

*a: When equivalence (no preference offered)*

*b: Forced choice.*

i. **Ratio of means (ROM)** is calculated after standardising individual ratios. An individual's initial preferred group is assigned a value of 1 and their less preferred group is assigned a value based on the number of people in their preferred group (note that this will be lower) divided by the number of people in the least preferred group (which will always be 100) at the point of indifference (using mid-point between the 2 group sizes within the final choice if they do not report equivalence). For example, if they select the equivalence option when Program A is 75 and Program B is 100 (hence they preferred Program A on the first choice), the value for Program A would be set to 1 and for Program B this would 75/100, or 0.75. The Ratio of Means is calculated by taking a mean of the assigned values for each Program across all respondents and reporting the ratio of mean values for Program A/Program B. Results will be calculated for each age group included and shown combined and separately based on whether the first question was forced or unforced. ROM will be reported both in table format and as graph, using age on the X axis.

ii. **Median of individual ratios (MOIR)** is based on producing each individual ratio then ordering all respondents' ratios and taking the middle ratio value. Results will be calculated for each age group included and shown combined and separately based on whether the first question was forced or unforced. MOIR will be reported both in table format and as graph, using age on the X axis.

ROM and MOIR for each age comparison will be compared and tested between 1) types of health and life expectancy gain and 2) the option of equivalence versus no option of equivalence. The additional consistency check question (100vs75) asked following an initial expression of 'no preference' when patient groups sizes are both 100 will not be included in the ROM or MOIR calculations.

We will explore the uncertainty around the ROM and MOIR using bootstrapping of the pairs of assigned values described above. We will use 1000 replications with replacement to calculate the percentile confidence interval (CI 2.5%, 97.5%). Other approaches will be explored depending upon the skew of the bootstrap data.

The data collected alongside the qualitative interviews will be combined with the main dataset.

Results of the test of chaining of responses will be considered by comparing the predicted equivalence value (from the chained PTOs e.g. age 2 vs age 12 and age 12 vs age 40) to the related actual equivalence value (e.g. age 2 vs age 40). The predicted (from chained PTO questions, see Table 1A) and actual values will be shown in a scatter plot. We will categorise respondents into meeting or not meeting the chaining test based on whether they display a preference reversal or if the estimate equivalence group size and the actual group size have a difference of 20 or more people (alternative thresholds will be considered). We will use logit regression to explore whether not meeting the chaining test is related to respondent characteristics or other indicators of data quality.

For each age comparison, we will report ratio of the means and median of the individual ratios, for the 5 health contexts. T-tests will be used to determine if the mean of the ratios is statistically significant different between two study arms (offering a choice of equivalence vs not offering the choice of equivalence), and Wilcoxon's U-test will be used to determine if median values are statistically significantly different between two arms. An example of how the data will be presented is shown in S4 Table. T-tests will be used to determine if the mean of the ratios value is statistically significantly different between two contexts comparing a health gain of 5 years versus 2 years, and Wilcoxon's U-test will be used to determine if median values are statistically significantly different. Kruskal-Wallis H tests will be used to determine if the ratios are statistically significantly different between the three contexts across relieving pain, mobility problems and emotional problems.

We will produce histograms showing the distribution of equivalence values for each age comparison; combined, and separately for the forced (dichotomous) and unforced (offering equivalence choice) modes.

Additionally, graphs showing the cumulative % of sample reporting equivalence at each equivalence (e.g. McNamara et al, 2021 [56]) will be presented. The X-axis will show 0 to 100 40-year-olds (or 50-year-olds) and then 100 to 0 younger ages (where X is either a single age or ages combined subject to ease of visual comprehension), the Y-axis will show the proportion of the full sample expressing indifference at each point. Graphs will be shown combined and disaggregated by forced and unforced initial responses (see Fig 10).

We will categorise respondents into seven classifications for each context according to their preferences as set out in Table 3.

Multinomial logistic regression analyses will be used to explore whether the individual's classification from their PTO responses (either based on Table 3 or simplified to three classifications) is associated with respondent characteristics and the age/context of the PTO question. This will treat the within childhood age comparisons (that are included in the chaining tests) separately to comparisons between children and adults.

Controls will include ages used in the PTO question, type of health gain, respondent demographic characteristics, and methods factors (whether the younger age appears on the left or right of the screen, forced or unforced preference). Alternative approaches to addressing the fact that individuals complete more than one PTO question will be explored including clustering standard errors and random effects multinomial logistic models.

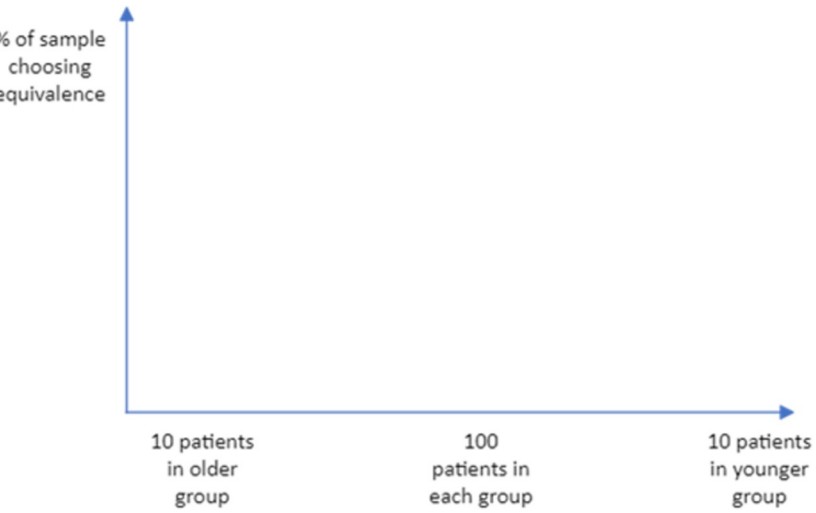

**Fig 10. Visual display of the percentage choosing equivalence at each group size comparison.**

**2.7.4 Comparing PTO values to attitudinal questions.** We will present the percentage of potentially inconsistent responders based on the rules in Table 4. These will be interpreted cautiously as there may be legitimate reasons for apparent contradictions.

**2.7.5 Reliability testing.** A standard means of exploring reliability of responses to survey questions would be to run a test-retest analysis on the same respondents (or a sample of the respondents) and report a correlation between respondents' first and second responses. However, because the number of patients in both age groups within the PTO questions can change (the least preferred group is held at 100), correlations between the PTO equivalence value responses (either number of patients or the ratio) in the first administration and the second administration are problematic. Respondents may report a different group is preferred in the second administration.

It would be possible to report the percentage of responses with the exact same equivalence answer for each question and a visual interpretation of the test-retest similarity through a scatter plot. However, this would not have a clear interpretation in terms of meeting any test-retest reliability criteria.

**Table 3. Preference classification based on PTO responses.**

| Classification | Threshold |
|---|---|
| Extreme preference for younger age group | Prefers to treat 1 younger age group than 100 older age group |
| Strongly prefer younger age group | Equivalence group size between 1 and 50 of younger age group equivalent to 100 of the older age group |
| Weakly prefer younger age group | Equivalence group size greater than 50 of younger age group equivalent to 100 older age group |
| Equal preference | Expresses equivalence when group sizes are the same or prefers to treat 100 of not preferred age group than 95 of the preferred age group |
| Weakly prefer older age group | Equivalence group size greater than 50 of older age group equivalent to 100 younger age group |
| Strongly prefer older age group | Equivalence group size between 1 and 50 of older age group equivalent to 100 of the younger age group |
| Extreme preference for older age group | Prefers to treat 1 older age group than 100 younger age group |

**Table 4. Identification of potentially inconsistent responders.**

| | Which of these statements best reflects your views? | Potentially inconsistent responses |
|---|---|---|
| 1 | For medical care that improves quality of life temporarily, children should have some priority over adults. | For any of the 3 quality of life questions relative weights suggests equal value or a pro-adult preference. |
| 2 | For medical care that improves quality of life temporarily, adults should have some priority over children. | For any of the 3 quality of life questions relative weights suggests equal value or a pro-child preference. |
| 3 | People should have the same priority with respect to medical care that improves quality of life temporarily, regardless of age. | For any of the 3 quality of life questions relative weights suggests a pro-adult or pro-child preference. |
| 4 | For medical care that extends life children should have some priority over adults. | For any of the life extension questions relative weights suggests equal value or a pro-adult preference. |
| 5 | For medical care that extends life adults should have some priority over children. | For any of the life extension questions relative weights suggests equal value or a pro-child preference. |
| 6 | People should have the same priority with respect to medical care that extends life regardless of age. | For any of the life extension questions relative weights suggests a pro-child or pro-adult preference. |
| 7 | This is fair because they are children | For any of the 7 PTO questions relative weights suggests equal value or a pro-adult preference. |
| 8 | This would make me feel concerned | For any of the 7 PTO questions relative weights suggests a pro-child preference. |
| 9 | This would be unfair | For any of the 7 PTO questions relative weights suggests pro-child preference. |

We will instead explore uncertainty of the main findings by reporting the bootstrap percentile confidence interval of the ratio of means and individual median of the ratios. From a decision maker perspective, the individual respondent consistency is less important than whether a replication of the study would generate consistent overall findings.

**2.7.6 Qualitative data analysis.** Analysis of the interviews will begin alongside data collection to enable lessons learnt and issues raised to feed into the interview process, potential amendments to the question prompts and considerations of whether saturation of themes has been reached. The analysis will adopt a thematic analysis approach drawing upon the steps outlined within framework analysis (Gale et al. 2013 [57]) (see S1 File).

## 2.8 Data access

At the completion of the project the anonymised survey data will be made available via Melbourne Figshare (melbourne.figshare.com).

## 2.9 Pilot testing and soft launch

**2.9.1 Input from QUOKKA's Consumer Advisory Group (CAG).** (https://www.quokkaresearchprogram.org/for-consumers-1).

The survey was discussed in its early development stages with the Consumer Advisory Group (CAG) for the overall QUOKKA programme of work. CAG members were supportive of the general approach we intended to use and provided useful feedback on simplification of the Plain Language Statement and the wording of the PTO survey introduction and choice tasks. CAG members raised concerns around including sensitive questions within focus groups, particularly those conducted online. They viewed some of the questions as potentially making respondents feel uncomfortable, particularly if discussed within a group of unknown

people. In response to their concerns, we amended aspects of the focus groups to concentrate the discussion on the decision maker perspective and use of study findings. The discussion around personal views on prioritising health gain based on age were limited to the one-to-one interviews.

Three members of the CAG also piloted the completed survey and provide feedback, particularly in relation to the practice exercise and the introduction to the survey.

**2.9.2 Pilots for the online survey.**   The survey was piloted on a sample of at least 14 convenience respondents in addition to the three CAG members noted above. These pilot surveys involved discussion with the respondents and considered the questions in Box 4, along with the respondent-reported structured feedback questions that form part of the survey.

Box 4. Prompts for the pilot of the online survey.

**General survey flow**:

- Did the respondent find anything in the survey distressing or inappropriate?

- Can the respondent navigate the survey in the expected way? Are there any points in the survey where the respondent asked for clarification?

**Interpreting the instructions on 'value'**:

- How did the respondent interpret the statement relating to the framing of the PTO questions and what they were requested to consider (e.g. they are told costs are the same for both Programs).

- Did they appear to follow these instructions throughout the survey?

**Interpreting PTO questions**:

- Did respondents interpret the PTO questions in the expected manner?

- Is the number of PTO questions for each respondent appropriate? Did they seem to be losing interest/getting tired at the end?

- Do respondents notice the changing health gain in the questions – length of life, emotional and physical health?

- Are the iterative questions appropriate – is 3 (or 4 for extreme preferences) iterative follow-on questions for each main question appropriate from their perspective?

- Where respondents reach the extreme option (e.g. 90/100, or 1/100) how close is this to their actual equivalence value?

**Feedback and attitudinal questions**:

- Are the comprehension, feedback and attitudinal questions easily understood?

- Did respondents feel that the multiple-choice options available expressed their views?

Data collected during the pilots will not be included in the full sample. Key feedback from the pilots and how the survey was amended is shown in S2 Table. The pilots were conducted by ADS and TP and included 8 males and 6 females between ages 14 and 79 years.

**2.9.3 Soft launch of the online data collection.** The online survey will collect a soft launch sample of 50 respondents. Further data collection will be paused at that point, while the initial data are analysed to consider:

- any unanticipated problems, particularly in relation to accessing and analysing the data files

- whether the unique survey URL links and landing pages are working appropriately

- whether the quota screening, randomization and survey flow is working appropriately

- whether the data aligns to the analysis code in the expected manner

- duration of survey completion

- responses to comprehension questions

- agreed quality control criteria for inclusion to the dataset and replacement by the recruiting company

If no substantive changes are required following this launch the data will be included within the full sample. If the soft launch raises concerns (e.g. low comprehension or randomization problems) the main data collection will be placed on hold until this is resolved, which may involve further pilot work.

**2.9.4 Pilot qualitative interview.** Two pilot qualitative interviews were conducted on a convenience sample (on people who do not have knowledge of health state valuation). These pilots were conducted by CB and included one male and one female aged 14 and 33. This data will not be included within the main qualitative sample. The pilot interviews were conducted to mirror the main qualitative interviews. However, at the end of the 'think aloud' and semi-structured interview section the respondents were also asked to give their feedback on the process of the 'think aloud'/interview component.

Consideration was given to:

- smoothness of conducting the interview, screen sharing, recording, and interacting with the online survey

- respondents' comprehension of the interview questions and prompts

- respondent feedback on the process and the survey

- potential for further probing to gain greater understanding of respondents thinking

Pilot interviews were recorded (via zoom) and re-watched to identify potential prompts which did not work well, and areas where additional questioning or different style of questioning may have been beneficial.

Pilot interviews will also be conducted with respondents recruited through CRNRStone (n = 6) by three interviewers TP, ADS, CB. These pilot interviews will be used to assess the quality of the interviews in terms of addressing the research questions and making necessary amendments prior to further interviews. Unless substantial changes are made to the interview prompts or survey following these pilots their data will be included in the final analysis.

**2.9.5 Pilot focus group and deliberative exercise.** One pilot focus group will be conducted on a convenience sample (on people who do not have knowledge of health state valuation). This data will not be included within the sample.

Consideration will be given to:

- respondent feedback on the process, the materials presented and the discussion
- potential for probing to gain greater understanding of respondents' views

## 2.10 Quality assurance

**2.10.1 Measures to support high quality data collection.** The study has adopted a number of approaches to support high quality data. To avoid order effects, we will randomise the question order where this does not impact on ease of comprehension and randomise the position on the screen (left or right) of the younger versus older age group. We will ensure the survey can be completed in about 20 minutes and is easy to understand and complete through careful piloting, initially with a convenience sample of friends and colleagues, then with the QUOKKA Consumer Advisory Panel. Piloting will include cognitive feedback on the PTO questions focusing on the feasibility and comprehension of the survey and are distinct from the qualitative interviews. In addition to piloting the survey, we will run a soft launch of the online data collection to ensure that data collection is performing as expected.

Inclusion to the main dataset will only be possible if respondents meet pre-set quality criteria which includes:

- not speeding (a minimum time of 25% of the median duration based on the soft launch will be applied in order to support replacement recruitment, this will be checked against the duration of the full sample during the analysis)

- responding correctly to an attention check question (*'This is an attention check question. Please select 'somewhat agree' from the following options*: *'agree, somewhat agree, neither agree nor disagree, somewhat disagree, disagree'*)

- reporting own age group consistently with own age which are asked at the start and end of the survey respectively

- not answering nonsense or rude text in the free text (optional) responses

- unique survey company ID number

- failing a 'honeypot' question designed only to be visible to bots.

**2.10.2 Tests of data quality.** The survey will build in a number of data quality checks including directly asking respondents about their comprehension of the questions. We will test whether seeing the young age group on the left of the screen results in different relative weights to seeing this on the right of the screen (via inclusion as a covariate in the multinomial regression on PTO preference classification see section 2.7.3). If screen position does impact on the results this will be interpreted as an indication of poor data quality.

## 2.11 Ethics

**2.11.1 Main ethical considerations.** Asking the public questions relating to health care prioritisation raises ethical issues as people must think about poor health, dying and potentially denying treatment to some groups. Completing PTO and attitudinal questions relating to health care prioritising may cause respondents distress, particularly those with, or having family with, serious health problems and/or potentially life limiting conditions. Potential respondents should be provided with a clear understanding of the study in order for them to make an informed decision about whether to take part. This is challenging without actually

working through the type of questions they will need to complete which would be unrealistic in the current recruitment set up. To minimise the risk of distress, the initial invite to the study will set clear expectations about what will be involved. For the one-to-one interviews and focus groups, interviewer training and an interview protocol will be followed to ensure that any respondents who become upset are treated appropriately.

There is a slight tension in presenting the value of the research both to ensure that respondents understand the motivation of the research and to encourage their participation whilst not overselling the likely contribution of the research to decision making.

Respondents include people aged 16 plus, across all demographics. We aim to ensure that no respondents are placed in a position in which they find their involvement alienating, confusing or embarrassing. Through careful testing, particularly with the Consumer Advisory Group, we have aimed for use of common language, and clear instructions. All respondents and their views will be treated with respect.

**2.11.2 Research ethics approval.**   The protocol outlined in this paper was granted ethical approval from the University of Melbourne human ethics committee [Reference number: 2023-24869-37630-4].

**2.11.3 Informed consent and confidentiality.**   The respondents will be sent information about the study in the form of a 'Plain Language Statement' (an example is shown in S5 File). In case participants do not read this, key information about the study is also shown at the start of the survey to ensure respondents are informed prior to giving consent.

Informed consent will be obtained at the start of the PTO survey for those undertaking the self-directed online survey. For those undertaking one-to-one interviews and focus groups, consent will be requested prior to the interview or focus group via a self-complete online questionnaire (on Qualtrics). If consent has not been recorded prior to the interview or focus group, it will be obtained prior to commencing the interview/focus group using the same online questionnaire.

The online self-directed PTO survey will not contain respondent's name or identifying data (e.g. postcode). Links to the self-complete respondents will only be possible via a unique URL link with an individual identification number provided by the survey company. The identification numbers of completed surveys will be communicated to the survey company. Respondents will be asked whether they would like to be sent a summary of the research findings; the identification numbers of those answering positively will also be sent to the survey company.

For the qualitative interviews, links to the respondents' data completed alongside the interview will only be possible via the data and time of the interview. Respondent personal details for the one-to-one interviews and focus groups will be recorded to facilitate recruitment and reimbursement; however, their quantitative and qualitative data will be identified only by study codes. Any personal details (name, email address) used in arranging interviews will be deleted at the end of recruitment. Names and email addresses of respondents requesting a summary of the research findings will be kept until the end of the project, and will be deleted after this is completed.

## 3. Discussion

### 3.1 Summary

This is a mixed methods study to provide evidence to decision makers in Australia on public opinion regarding the social value of child health gains relative to adult health gains. Specifically, we aim to provide an estimate of the average relative weight for child health gains relative to adult health gains as judged by the Australian general public based on responses to an online PTO study.

### 3.2 Strengths of the planned study

The PTO survey will provide relative weights for 13 different child and young person ages relative to adults of two different ages, sufficient to provide identification of non-linearities in relative social value of children and young people. The inclusion of PTO questions on extending length of life and improving quality of life of different dimensions within the same study will enable a direct comparison of the impact of context on respondent preferences. The sample size of the study is sufficient to include a methodological test of the impact of forcing choices in the PTO question or allowing respondents to express equal preference for the different age groups. This understanding will be valuable for future related studies.

The use of mixed methods with in-depth cognitive and 'think aloud' interviews will support our understanding of the motivations for responses and help understand inconsistencies between PTO style and attitudinal style questions that have been identified in the literature. The survey has been rigorously tested and piloted prior to launch. This has included input from the QUOKKA Consumer Advisory Group, who have been able to improve the clarity and comprehension of the questions and to ensure sensitive questions are asked in an appropriate way.

### 3.3 Limitations of the planned study

The use of online data without an interviewer present for the majority of the sample assumes that respondents are willing to concentrate and engage in the study. The recruitment company chosen for the study has a recruitment and reward approach which mitigates against some common concerns around use of commercial online recruitment. In addition, quality control procedures should help to ensure only genuine responses are included. Although quota sampling will be used to ensure representation on observable characteristics such as education and age, it is acknowledged that recruitment through commercial marketing companies, and volunteering to participate in academic research, may not capture all social groups.

Limitations of the PTO design were discussed in Section 2.1.3. Where possible we have aimed to counter these limitations (e.g. random ordering of questions to address the risk of ordering effects, holding the least preferred group at 100 and offering iterative questions up to 1 versus 100 patients to identify extreme preferences versus lack of willingness to trade and limit the likely skew of the data) and incorporate a means to explore those potential limitations (e.g. comparison of PTO to attitudinal questions to help draw the distinction between respondent views on distributional principles versus preferences when faced with choice based questions involving numbers of patients).

The study is limited to collecting data in Australia therefore may not be generalisable to other countries.

## 4. Conclusion

This study should provide valuable information to health care decision makers on the Australian public views of assigning additional weight to health gains for young people. It will also contribute to available evidence on the reasoning of respondents and to a better understanding of the use of PTO surveys to elicit health care preferences.

## Supporting information

**S1 Table. Maximum sampling quotas for the ORU online sample.**
(DOCX)

**S2 Table. Issues raised with the survey pilots and how they were addressed.**
(DOCX)

**S3 Table. Survey attributes.**
(DOCX)

**S4 Table. Comparison of forced versus unforced choice by health context and age comparison.**
(DOCX)

**S1 File. Analysis steps adopting a thematic approach.**
(DOCX)

**S2 File. Interview prompts.**
(DOCX)

**S3 File. Interview protocol and risk management.**
(DOCX)

**S4 File. Focus group prompts.**
(DOCX)

**S5 File. Example plain language statement.**
(DOCX)

## Acknowledgments

The authors wish to acknowledge with thanks helpful feedback and advice received on the conception and design of this study by QUOKKA's two principal advisory groups: our Decision Makers' Panel and our Consumer Advisory Group. We are also grateful to the Commonwealth Department of Health, Australia, for valuable guidance to the QUOKKA research program. We are also grateful to Lea Kevin-Tidis who provided administrative support and liaison. We would also like to thank colleagues, friends and family who gave up their time to complete pilot interviews and surveys.

Koonal Shah is an employee of the National Institute for Health and Care Excellence. However, the views expressed in this paper are not necessarily the views of the National Institute for Health and Care Excellence.

## Author Contributions

**Conceptualization:** Tessa Peasgood, Cate Bailey, Gang Chen, Ashwini De Silva, Richard Norman, Koonal Shah, Rosalie Viney, Nancy Devlin.

**Data curation:** Tessa Peasgood, Cate Bailey, Ashwini De Silva, Udeni De Silva Perera.

**Formal analysis:** Tessa Peasgood, Cate Bailey, Gang Chen, Ashwini De Silva, Udeni De Silva Perera, Nancy Devlin.

**Funding acquisition:** Tessa Peasgood, Gang Chen, Richard Norman, Rosalie Viney, Nancy Devlin.

**Investigation:** Tessa Peasgood, Cate Bailey, Gang Chen, Ashwini De Silva, Udeni De Silva Perera, Nancy Devlin.

**Methodology:** Tessa Peasgood, Cate Bailey, Gang Chen, Ashwini De Silva, Udeni De Silva Perera, Richard Norman, Koonal Shah, Rosalie Viney, Nancy Devlin.

**Project administration:** Tessa Peasgood, Cate Bailey, Ashwini De Silva.

**Supervision:** Tessa Peasgood, Gang Chen, Nancy Devlin.

**Writing – original draft:** Tessa Peasgood.

**Writing – review & editing:** Tessa Peasgood, Cate Bailey, Gang Chen, Ashwini De Silva, Udeni De Silva Perera, Richard Norman, Koonal Shah, Rosalie Viney, Nancy Devlin.

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
