## [Decision Letter · Decision Letter 0]

20 Oct 2023

PONE-D-23-14388Protocol for a mixed methods Person Trade Off (PTO) and qualitative study to estimate and understand the relative value of gains in health for children compared to adultsPLOS ONE

Dear Dr. Peasgood,  

Thank you for submitting your manuscript to PLOS ONE. After careful consideration, we feel that it has merit but does not fully meet PLOS ONE’s publication criteria as it currently stands. Therefore, we invite you to submit a revised version of the manuscript that addresses the points raised during the review process.

We look forward to receiving your revised manuscript.

Kind regards,

Muhammad Anwar, PHD

Academic Editor

PLOS ONE

Journal Requirements:

"I have read the journal's policy and the authors of this manuscript have the following competing interests: [ND, RV, RN, KS and TP are all members of the EuroQol group]"

4. Please upload a new copy of Figures 1a, 1b, 1c, 1d, 3, and 5  as the detail is not clear. Please follow the link for more information: " ext-link-type="uri" xlink:type="simple">https://blogs.plos.org/plos/2019/06/looking-good-tips-for-creating-your-plos-figures-graphics/"
https://blogs.plos.org/plos/2019/06/looking-good-tips-for-creating-your-plos-figures-graphics

Reviewers' comments:

Reviewer's Responses to Questions

**Comments to the Author**

1. Does the manuscript provide a valid rationale for the proposed study, with clearly identified and justified research questions?

Reviewer #1: Partly

Reviewer #2: Yes

2. Is the protocol technically sound and planned in a manner that will lead to a meaningful outcome and allow testing the stated hypotheses?

Reviewer #1: Partly

Reviewer #2: Yes

3. Is the methodology feasible and described in sufficient detail to allow the work to be replicable?

Reviewer #1: No

Reviewer #2: Yes

4. Have the authors described where all data underlying the findings will be made available when the study is complete?

Reviewer #1: No

Reviewer #2: Yes

5. Is the manuscript presented in an intelligible fashion and written in standard English?

Reviewer #1: Yes

Reviewer #2: Yes

6. Review Comments to the Author

You may also provide optional suggestions and comments to authors that they might find helpful in planning their study.

Reviewer #1: 1. The title, though linked ot the funding objective, is misleading as it implies a balance between children, and adults as a homogenous concept. It would be ore accurate to refer to 'childhood and alter stages of life'.

2. The Introduction cites only English speaking, historically Christian and formerly British, contexts. This will limit thee general scientific utility of the results if the majority of global cultures and values are overlooked (but recognising that the locus of the study is Australia).

3. The European network EUNETHA (see www.eunetha.eu) is not mentioned, but this brings together numerous institutions across over 20 countries to harmonise methodologies.

4. An underlying duality of concepts through the paper is extension of life by two years, and a time-limited period of illness with no lasting outcomes. This seems to unduly simplify the conundrum being studied, possibly to the extent of rendering it of limited utility, as many real world decisions will be between treatments with differing levels of reduced quality of life over a long period; or between a short good quality period of life and a longer but poorer quality of life (for example 2 good quality years of life vs 5 poorer quality of life years). Respondents' views may differ on these options according to patient age.

5. Presentation of the paper is very discursive, with protocol components emerging within this process. A much crisper means of presenting Issue - Evaluation of Factors/Literature - Decision for this Protocol, should be devised.

6. It is of concern that the sampling vehicle and methodology have not yet been decided. This will be a challenge, but will be crucial to the work and so options should be more fully appraised in this protocol.

7. Given the theoretical and conceptual nature of the questions, and indeed to locus of 'national policy maker', there is a great danger of upper middle class bias in the study participation (the product of sampling and subsequent positive response). How will this be mitigated? All components of society should be able to feel equally involved, such as bus drivers, shop workers, agricultural workers, and the lower-educated retired.

8. The choices in section 2.3.5 seem limited. 'Impairment in activities of normal living' (appropriate to age) would seem an important measure of reduced quality of life.

9. Demographics of respondents seem framed in age, sex, and education. Surely employment type, income level, and a measure of neighbourhood type or housing type, are important in looking at how real society is represented.

10. There is no mention of piloting, which is essential. Moreover, a comprehension analysis of both the video (which is quite academic) and the study tools, is desirable.

11. In assessing potential respondent values or bias, a question asking if any of the examples (specify which) exist or have occurred in the respondent's family or close contacts would be illuminating.

12. It appears that the data will be transferred dynamically to Utah, USA. There should be comment on the data protection issues of this within the context of Australian law.

Reviewer #2: This is clearly written and very thorough overall. Sufficient detail is given at each stage. In fact, the paper seems quite long to me but I’m assuming that it conforms to journal requirements in this regard. Otherwise, I have only minor comments, detailed below.

In the abstract the phrase ‘similar cultural background’ struck me as being a little crude, but I can’t think of a better one!

p.6 ‘set out the catalyst for undertaking this work’. I’m not sure if the paper really describes a ‘catalyst’? maybe ‘rationale’ would serve better?

There is commendable detail on the rationale behind each step of the proposed research. The in-depth consideration of alternative designs (section 2.1) is interesting but perhaps not really required in this level of depth in a protocol paper. But if word length is not an issue there’s no harm in keeping this in. similarly, I’m not used to seeing a ‘discussion’ section in a protocol paper, albeit this is fairly brief in this case.

p. 12 focus group sample size ‘The size of the focus groups will be kept small (=5 participants) to encourage full participation and discussion’. Okay, but generating good discussion is often helped by having a larger group (e.g. 8-10) and this also mitigates against the risk of low attendance on the day. I’m not sure going down to potentially 2 participants is ideal. However, this is an experienced team so if this is their judgement, fair enough.

p.12 add word ‘in’ to sentence ‘This approach places the primary trade off we are interested IN as the main focus of the respondents’ attention’

p.36 box 4 change ‘Australian governments’ to ‘Australian government’

p.39 2.4.1 – no need for question marks after points i and ii.

the youtube clip was excellent (p.16)

p. 47 think it should be respondents ‘who’ report equivalence (not ‘whom’)

some of the figures haven’t come out well on the PDF I downloaded

Use of the ampersand in headers and subheaders is currently a bit random (e.g. see 2.3.12 and 2.3.13)

p.56 if you are adopting the framework approach (Gale) I’m not sure you need a reference to Braun and Clarke (whose approach is a little different)

7. PLOS authors have the option to publish the peer review history of their article (what does this mean?). If published, this will include your full peer review and any attached files.

Reviewer #1: No

Reviewer #2: No

---

## [Author Response · Author response to Decision Letter 0]

18 Jan 2024

We would like to sincerely thank the reviewers for taking the time to assess our manuscript. We have carefully considered all their comments. Below we provide the point-by-point responses. All quotes from the manuscript have been shown in italics.

Responses to Reviewer #1: 

1. The title, though linked to the funding objective, is misleading as it implies a balance between children, and adults as a homogenous concept. It would be more accurate to refer to 'childhood and alter stages of life'. 

We agree the title may seem misleading. We have amended the title to refer to ‘children and young people’ which more accurately portrays the research since we will compare young people up to age 24 to adults (aged 40 or 55).

2. The Introduction cites only English speaking, historically Christian and formerly British, contexts. This will limit the general scientific utility of the results if the majority of global cultures and values are overlooked (but recognising that the locus of the study is Australia).

The reviewer raises a good point about the generalisability of the findings. The intention of this work, and that of the funders, was to generate evidence to support decision making in the Australian health care context only. However, we hope that both the findings and the testing of methods within the study will be of interest to international audiences. Additionally, we hope that this detailed protocol may be of value to international researchers who may wish to conduct similar/replication work in other countries. 

To acknowledge the limits to generalisability we have included an additional statement in the limitations section, which states:

“Our study is limited to collecting data in Australia (in English only) therefore may not be generalisable to other countries”

Additionally, we will ensure a similar caveat is included in any subsequent reports of the study.

3. The European network EUNETHA (see www.eunetha.eu) is not mentioned, but this brings together numerous institutions across over 20 countries to harmonise methodologies.

Thank you - we have added a reference to the main EUnetHTA core model which is a useful reference for the reader in terms of understanding the scope of HTA and also helps set HTA in an international context beyond Australia.

4. An underlying duality of concepts through the paper is extension of life by two years, and a time-limited period of illness with no lasting outcomes. This seems to unduly simplify the conundrum being studied, possibly to the extent of rendering it of limited utility, as many real world decisions will be between treatments with differing levels of reduced quality of life over a long period; or between a short good quality period of life and a longer but poorer quality of life (for example 2 good quality years of life vs 5 poorer quality of life years). Respondents' views may differ on these options according to patient age.

We agree that respondent views may differ depending upon the quality of life of any additional years. We appreciate that focusing on years of life in good health is a simplification - however after conducting a systematic review of the evidence on child/adult weights (published in Pharmacoeconomics, https://link.springer.com/article/10.1007/s40273-023-01327-x) we considered that there remained an important gap in the knowledge base on understanding different attitudes to prioritising children for years of life extension compared to quality of life when treated separately which was something we wanted to explore. There was also a gap in understanding the impact of different types of quality of life (e.g. pain, anxiety, limitations in functioning) which we also wanted to explore. 

In the study we were also interested in a number of further considerations including: investigating a broad range of ages from baby to young adult, different numbers of years of life extension, and methodological differences through offering a forced choice or not. In order to ensure a reasonable sample size for each comparison (within a fixed budget) we could not include all considerations and unfortunately questions around including additional years of life spent in poor health were judged as less of a research priority in the current study. This has been added as a limitation.

5. Presentation of the paper is very discursive, with protocol components emerging within this process. A much crisper means of presenting Issue - Evaluation of Factors/Literature - Decision for this Protocol, should be devised.

We have edited the paper to attain greater clarity, removed some of the repetition and have aimed for a more concise structure with a reduced word count. 

We do also acknowledge that in addition to aspects of the planned study design, we have also included discussion around methods choices. This has been included as we felt it would be of value to researchers and HTA bodies wishing to conduct or commission future research in this, or related, areas. For example, NICE introduced a severity modifier to their HTA methods guidance in 2022 with the intention to conduct future research into the appropriate magnitude of this modifier and to gain a better understanding of societal preferences (Angelis et al, 2023). We aimed to make this protocol of value to future researchers exploring social preferences across attributes beyond age weights. 

Angelis A, Harker M, Cairns J, Seo MK, Legood R, Miners A, Wiseman V, Chalkidou K, Grieve R, Briggs A. The evolving nature of Health Technology Assessment: a critical appraisal of NICE’s new methods manual. Value in Health. 2023 Jun 1.

6. It is of concern that the sampling vehicle and methodology have not yet been decided. This will be a challenge, but will be crucial to the work and so options should be more fully appraised in this protocol.

We have added some additional clarity on use of quotas relating to age/sex groups, education qualification attained, and location in terms of state/territory based on the Australian census (see section 2.6.2).

This now reads;

“Members of the ORU panel aged 16 and above will be invited to complete the survey. Drawing upon 2021 Census data, the sampling quotas will adopt broadly equal male and female recruitment in each of seven age-gender groups, with a slight over-sampling opportunity of about 5% in each of the age-gender groups to avoid excessive rejection of respondents at screening. Additionally, a quota will be applied based on attainment of a degree to ensure a balance of education levels which can also proxy for other socio-economic indicators. Loose geographical quotas will also be applied to give a reasonable state/territory mix. Respondents will complete a screening questionnaire and will only be able to enter the survey if their age-gender, education and location quotas have not yet been reached (see S1 Table for quotas). As our main concern is the ability to explore heterogeneity of preferences between parents and non-parents, it was not considered necessary to quota sample based on parent status. The data collection will be paused at a mid-point to explore the sample in relation to parenthood status and consider whether additional quotas are required for the remaining data collection.”

7. Given the theoretical and conceptual nature of the questions, and indeed to locus of 'national policy maker', there is a great danger of upper middle class bias in the study participation (the product of sampling and subsequent positive response). How will this be mitigated? All components of society should be able to feel equally involved, such as bus drivers, shop workers, agricultural workers, and the lower-educated retired.

We acknowledge this concern. The sampling quotas for education help generate a balanced sample in terms of education but we accept this may not go far enough to ensure fully diverse recruitment or reach marginalised groups. To acknowledge this we have added the following limitation:

“Although quota sampling will be used to ensure representation on observable characteristics such as education and age, it is acknowledged that recruitment through commercial marketing companies, and volunteering to participate in academic research may not capture all social groups.”

Additionally, we will ensure a similar caveat is included into any future write up of the study.

8. The choices in section 2.3.5 seem limited. 'Impairment in activities of normal living' (appropriate to age) would seem an important measure of reduced quality of life.

We agree that there are other key health decrements which could have been used, including impairment of activities of normal living. However, given the limitation in total sample size we opted to focus on just three core aspects. These were chosen in part to link to dimensions in the EQ-5D-Y measure of health-related quality of life (mobility, pain, anxiety/depression). We wished to include both a mental health and a physical health decrement as we considered that to be an important comparison. We also wished to include an aspect of physical health which could be seen as having a very similar experience despite patient age (i.e. pain) and one which could be considered to have a slightly different impact (i.e. mobility) but was still able to be described such that it was the same health decrement that was being compared. Whilst decrements in usual activities or activities of normal living would be interesting, respondents are unlikely to be comparing the same thing across the life course (of course that is also what makes it interesting, but it is not the focus of this study which is trying to isolate the age-effect).

9. Demographics of respondents seem framed in age, sex, and education. Surely employment type, income level, and a measure of neighbourhood type or housing type, are important in looking at how real society is represented.

Yes - the sampling could have included other issues. However, we thought age was the most important given the topic and we therefore set combined age and gender crosstab quotas. In the systematic review (published in Pharmacoeconomics; https://link.springer.com/article/10.1007/s40273-023-01327-x) we found little evidence on the individual characteristics that are associated with views on age-weighting health gains beyond age and gender. We also want to limit the number of rejections for respondents completing screening questionnaires through sampling on many characteristics. 

The sampling strategy was discussed with decision makers (QUOKKAs Decision Maker Panel) and their main concerns related to representation based on age and that recruitment included 16 and 17 year olds. 

Although the quota sampling is limited to age, gender, education, and location we will also report parenthood status (including grandparent status), and ethnicity (being born in Australia and speaking a language other than English at home) to assess the representativeness of the sample on these characteristics (see 2.3.14).

10. There is no mention of piloting, which is essential. Moreover, a comprehension analysis of both the video (which is quite academic) and the study tools, is desirable.

Piloting was conducted on the survey based on a convenience sample of 14 responders for the survey (from ages 14 to 79) and two respondents for the one-to-one interview. The QUOKKA project consumer advisory group (which is composed of parents) also discussed the project and survey in various draft stages during advisory group meetings. Additionally, three individuals from the consumer advisory group formally piloted the survey along with a cognitive debrief. The consumer advisory group members helped to improve the clarity of the language in the survey and the introduction to the survey, including the video. 

The piloting is discussed in section 2.9. Piloting and amendments were undertaken iteratively. Following revisions during the pilot (which are documented in Supplementary Table S2), we found that the final version of the survey was well understood by later pilot respondents. 

11. In assessing potential respondent values or bias, a question asking if any of the examples (specify which) exist or have occurred in the respondent's family or close contacts would be illuminating.

We did consider doing this. However, we were very conscious of the length of the survey and the need to ask quite a lot of questions about a respondent’s family in order to make this question useful for analysis (e.g. what health condition, which family member, when did it occur). In order to keep the survey concise, and not collect personal health data which we were uncertain would be of value, we ask only about whether the respondent themselves and their children have experienced a serious health condition.

12. It appears that the data will be transferred dynamically to Utah, USA. There should be comment on the data protection issues of this within the context of Australian law.

Apologies, this was not clear. The head office for Qualtrics is based in Utah and this was added as the reference to the Qualtrics platform. However, when the data is collected via the University of Melbourne’s use of the Qualtrics platform the data is held at Amazon AWS Data Centres in Sydney. To avoid confusion the reference to Utah has been removed. 

Responses to Reviewer #2: 

This is clearly written and very thorough overall. Sufficient detail is given at each stage. In fact, the paper seems quite long to me but I’m assuming that it conforms to journal requirements in this regard. Otherwise, I have only minor comments, detailed below.

Thank you for the supportive comments. We have edited the protocol slightly to reduce the length and reduced the word count by about 7%. 

In the abstract the phrase ‘similar cultural background’ struck me as being a little crude, but I can’t think of a better one!

We have removed this statement - the primary intention is to provide evidence for the Australian context. The sentence now reads: 

“Our planned study will provide valuable information to healthcare decision makers in Australia who may need to decide whether to pay more for health gains for children and young people compared with adults.”

p.6 ‘set out the catalyst for undertaking this work’. I’m not sure if the paper really describes a ‘catalyst’? maybe ‘rationale’ would serve better?

Thank you - we agree your suggestion is better so have implemented this change from ‘catalyst’ to ‘rationale’. 

There is commendable detail on the rationale behind each step of the proposed research. The in-depth consideration of alternative designs (section 2.1) is interesting but perhaps not really required in this level of depth in a protocol paper. But if word length is not an issue there’s no harm in keeping this in. similarly, I’m not used to seeing a ‘discussion’ section in a protocol paper, albeit this is fairly brief in this case.

We appreciate that usually a protocol would not contain this level of discussion around design issues. We were keen to be transparent around the options and issues considered behind the design choices. Publishing in PLOS One allows for this flexibility. 

HTA agencies are continuing to explore the methods for eliciting public preferences around social value. The considerations around methods for elicitation of age-weights are also relevant to other aspects of social value and timely (e.g. NICE’s severity modifier, included in the 2022 methods guidance, will be subject to further research to explore its magnitude and social preferences for QALY weights (Angelis et al, 2023)). We felt our discussion around methods considerations could make a valuable contribution to this current debate and future research. 

Angelis A, Harker M, Cairns J, Seo MK, Legood R, Miners A, Wiseman V, Chalkidou K, Grieve R, Briggs A. The evolving nature of Health Technology Assessment: a critical appraisal of NICE’s new methods manual. Value in Health. 2023 Jun 1.

p. 12 focus group sample size ‘The size of the focus groups will be kept small (=5 participants) to encourage full participation and discussion’. Okay, but generating good discussion is often helped by having a larger group (e.g. 8-10) and this also mitigates against the risk of low attendance on the day. I’m not sure g

---

## [Decision Letter · Decision Letter 1]

16 Feb 2024

PONE-D-23-14388R1Protocol for a mixed methods Person Trade Off (PTO) and qualitative study to estimate and understand the relative value of gains in health for children and young people compared to adultsPLOS ONE

Dear Dr. Peasgood,

Thank you for submitting your manuscript to PLOS ONE. After careful consideration, we feel that it has merit but does not fully meet PLOS ONE’s publication criteria as it currently stands. Therefore, we invite you to submit a revised version of the manuscript that addresses the points raised during the review process.

We look forward to receiving your revised manuscript.

Kind regards,

Muhammad Anwar, PHD

Academic Editor

PLOS ONE

Journal Requirements:

Reviewers' comments:

Reviewer's Responses to Questions

**Comments to the Author**

1. Does the manuscript provide a valid rationale for the proposed study, with clearly identified and justified research questions?

Reviewer #1: Yes

Reviewer #2: Yes

2. Is the protocol technically sound and planned in a manner that will lead to a meaningful outcome and allow testing the stated hypotheses?

Reviewer #1: Yes

Reviewer #2: Yes

3. Is the methodology feasible and described in sufficient detail to allow the work to be replicable?

Reviewer #1: Yes

Reviewer #2: Yes

4. Have the authors described where all data underlying the findings will be made available when the study is complete?

Reviewer #1: Yes

Reviewer #2: Yes

5. Is the manuscript presented in an intelligible fashion and written in standard English?

Reviewer #1: Yes

Reviewer #2: Yes

6. Review Comments to the Author

You may also provide optional suggestions and comments to authors that they might find helpful in planning their study.

Reviewer #1: This is a significant proposal, and the paper is now much clearer and more accessible. Thank you for making these changes. However, the paper is still much fuller than a traditional study protocol, and the extent of your analysis is hidden by the presentation and in particular the title. Conversely, though the title emphasises the PTO approach, this is only introduced quite surreptitiously in section 2.3.

The paper would be more appropriately titled: "Rationale, conceptual issues, and resultant Person Trade Off (PTO) protocol for a mixed methods and qualitative study to estimate and understand the relative value of gains in health for children and young people compared to adults".

In this vein, the key words should be reconsidered and re-ordered.

Reviewer #2: My comments have all been satisfactorily explained/addressed. on this basis i am happy to recommend publication

7. PLOS authors have the option to publish the peer review history of their article (what does this mean?). If published, this will include your full peer review and any attached files.

Reviewer #1: No

Reviewer #2: **Yes: **Iestyn Williams

---

## [Author Response · Author response to Decision Letter 1]

19 Feb 2024

We would like to sincerely thank the reviewers for taking the time to re-assess our manuscript and evaluate the changes made. Below we address the remaining concerns.

Reviewer #1: This is a significant proposal, and the paper is now much clearer and more accessible. Thank you for making these changes. However, the paper is still much fuller than a traditional study protocol, and the extent of your analysis is hidden by the presentation and in particular the title. Conversely, though the title emphasises the PTO approach, this is only introduced quite surreptitiously in section 2.3.

The paper would be more appropriately titled: "Rationale, conceptual issues, and resultant Person Trade Off (PTO) protocol for a mixed methods and qualitative study to estimate and understand the relative value of gains in health for children and young people compared to adults".

Response: We agree that your suggested title more accurately describes the content of the paper and is superior to the current one. We have therefore amended the title in line with your suggestion. Thank you.

In this vein, the key words should be reconsidered and re-ordered.

Response: Thank you for noting this. The current choice of key words are: “Person Trade Off (PTO); valuation health; children; Mixed Methods; social value; Australia; qualitative; online survey; public preferences” We have now amended the order to give less prominence to PTO and included a couple of more general terms to communicate the discussion components included in this protocol.

The key words are now (with additional terms underlined):

 “Social value; priority setting; public preferences; Person Trade Off (PTO); valuation health; children; mixed methods; qualitative; online survey; conceptual issues; Australia”

---

## [Decision Letter · Decision Letter 2]

16 Apr 2024

Rational, conceptual issues and resultant protocol for a mixed methods Person Trade Off (PTO) and qualitative study to estimate and understand the relative value of gains in health for children and young people compared to adults

PONE-D-23-14388R2

Dear Dr. Tessa Peasgood,

We’re pleased to inform you that your manuscript has been judged scientifically suitable for publication and will be formally accepted for publication once it meets all outstanding technical requirements.

Kind regards,

Muhammad Anwar, PHD

Academic Editor

PLOS ONE

Additional Editor Comments (optional):

Reviewers' comments:

Reviewer's Responses to Questions

**Comments to the Author**

1. Does the manuscript provide a valid rationale for the proposed study, with clearly identified and justified research questions?

Reviewer #1: Yes

2. Is the protocol technically sound and planned in a manner that will lead to a meaningful outcome and allow testing the stated hypotheses?

Reviewer #1: Yes

3. Is the methodology feasible and described in sufficient detail to allow the work to be replicable?

Reviewer #1: Yes

4. Have the authors described where all data underlying the findings will be made available when the study is complete?

Reviewer #1: Yes

5. Is the manuscript presented in an intelligible fashion and written in standard English?

Reviewer #1: Yes

6. Review Comments to the Author

You may also provide optional suggestions and comments to authors that they might find helpful in planning their study.

Reviewer #1: Thank you for your constructive response to review comments.

7. PLOS authors have the option to publish the peer review history of their article (what does this mean?). If published, this will include your full peer review and any attached files.

Reviewer #1: No

---

## [Editor Report · Acceptance letter]

8 May 2024

PONE-D-23-14388R2 

PLOS ONE

Dear Dr. Peasgood, 

I'm pleased to inform you that your manuscript has been deemed suitable for publication in PLOS ONE. Congratulations! Your manuscript is now being handed over to our production team.

Kind regards, 

on behalf of

Dr. Muhammad Anwar 

Academic Editor

PLOS ONE